# MODEL-BASED OFFLINE META-REINFORCEMENT LEARNING WITH REGULARIZATION

**Sen Lin**[1], **Jialin Wan**[1], **Tengyu Xu**[2], **Yingbin Liang**[2], **Junshan Zhang**[1,3]
[1]School of ECEE, Arizona State University  [2]Department of ECE, The Ohio State University
[3]Department of ECE, University of California, Davis
{slin70, jwan20}@asu.edu, {xu.3260, liang.889}@osu.edu, jazh@ucdavis.edu

## ABSTRACT

Existing offline reinforcement learning (RL) methods face a few major challenges, particularly the distributional shift between the learned policy and the behavior policy. Offline Meta-RL is emerging as a promising approach to address these challenges, aiming to learn an informative meta-policy from a collection of tasks. Nevertheless, as shown in our empirical studies, offline Meta-RL could be outperformed by offline single-task RL methods on tasks with good quality of datasets, indicating that a right balance has to be delicately calibrated between "exploring" the out-of-distribution state-actions by following the meta-policy and "exploiting" the offline dataset by staying close to the behavior policy. Motivated by such empirical analysis, we propose model-based offline Meta-RL with regularized Policy Optimization (MerPO), which learns a meta-model for efficient task structure inference and an informative meta-policy for safe exploration of out-of-distribution state-actions. In particular, we devise a new meta-Regularized model-based Actor-Critic (RAC) method for within-task policy optimization, as a key building block of MerPO, using both conservative policy evaluation and regularized policy improvement; and the intrinsic tradeoff therein is achieved via striking the right balance between two regularizers, one based on the behavior policy and the other on the meta-policy. We theoretically show that the learnt policy offers guaranteed improvement over both the behavior policy and the meta-policy, thus ensuring the performance improvement on new tasks via offline Meta-RL. Experiments corroborate the superior performance of MerPO over existing offline Meta-RL methods.

## 1 INTRODUCTION

Offline reinforcement learning (a.k.a., batch RL) has recently attracted extensive attention by learning from offline datasets previously collected via some behavior policy (Kumar et al., 2020). However, the performance of existing offline RL methods could degrade significantly due to the following issues: 1) the possibly poor quality of offline datasets (Levine et al., 2020) and 2) the inability to generalize to different environments (Li et al., 2020b). To tackle these challenges, offline **Meta**-RL (Li et al., 2020a; Dorfman & Tamar, 2020; Mitchell et al., 2020; Li et al., 2020b) has emerged very recently by leveraging the knowledge of similar offline RL tasks (Yu et al., 2021a). The main aim of these studies is to enable *quick policy adaptation* for new offline tasks, by learning a meta-policy with robust task structure inference that captures the structural properties across training tasks.

Because tasks are trained on offline datasets, value *overestimation* (Fujimoto et al., 2019) inevitably occurs in dynamic programming based offline Meta-RL, resulted from the distribution shift between the behavior policy and the learnt task-specific policy. To guarantee the learning performance on new offline tasks, a right balance has to be carefully calibrated between "exploring" the out-of-distribution state-actions by following the meta-policy, and "exploiting" the offline dataset by staying close to the behavior policy.

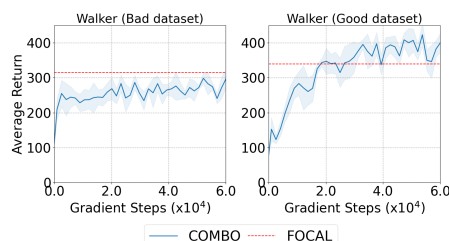

Figure 1: FOCAL vs. COMBO.

However, such a unique "exploration-exploitation" tradeoff has not been considered in existing offline Meta-RL approaches, which would likely limit their ability to handle diverse offline datasets

particularly towards those with good behavior policies. To illustrate this issue more concretely, we compare the performance between a state-of-the-art **offline Meta-RL** algorithm FOCAL (Li et al., 2020b) and an **offline single-task RL** method COMBO (Yu et al., 2021b) in two new offline tasks. As illustrated in Figure 1, while FOCAL performs better than COMBO on the task with a bad-quality dataset (left plot in Figure 1), it is outperformed by COMBO on the task with a good-quality dataset (right plot in Figure 1). Clearly, existing offline Meta-RL fails in several standard environments (see Figure 1 and Figure 11) to generalize universally well over datasets with varied quality. In order to fill such a substantial gap, we seek to answer the following key question in offline Meta-RL:

*How to design an efficient offline Meta-RL algorithm to strike the right balance between exploring with the meta-policy and exploiting the offline dataset?*

To this end, we propose MerPO, a model-based offline **Me**ta-RL approach with **r**egularized **P**olicy **O**ptimization, which learns a meta-model for efficient task structure inference and an informative meta-policy for safe exploration of out-of-distribution state-actions. Compared to existing approaches, MerPO achieves: (1) *safe policy improvement*: performance improvement can be guaranteed for offline tasks regardless of the quality of the dataset, by strike the right balance between exploring with the meta-policy and exploiting the offline dataset; and (2) *better generalization capability*: through a conservative utilization of the learnt model to generate synthetic data, MerPO aligns well with a recently emerging trend in supervised meta-learning to improve the generalization ability by augmenting the tasks with "more data" (Rajendran et al., 2020; Yao et al., 2021). Our main contributions can be summarized as follows:

(1) Learnt dynamics models not only serve as a natural remedy for task structure inference in offline Meta-RL, but also facilitate better exploration of out-of-distribution state-actions by generating synthetic rollouts. With this insight, we develop a model-based approach, where an offline meta-model is learnt to enable efficient task model learning for each offline task. More importantly, we propose a meta-regularized model-based actor-critic method (RAC) for within-task policy optimization, where a novel regularized policy improvement module is devised to calibrate the unique "exploration-exploitation" tradeoff by using an interpolation between two regularizers, one based on the behavior policy and the other on the meta-policy. Intuitively, RAC generalizes COMBO to the multi-task setting, with introduction of a novel regularized policy improvement module to strike a right balance between the impacts of the meta-policy and the behavior policy.

(2) We theoretically show that under mild conditions, the learnt task-specific policy based on MerPO offers safe performance improvement *over both the behavior policy and the meta-policy* with high probability. Our results also provide a guidance for the algorithm design in terms of how to appropriately select the weights in the interpolation, such that the performance improvement can be guaranteed for new offline RL tasks.

(3) We conduct extensive experiments to evaluate the performance of MerPO. More specifically, the experiments clearly show the safe policy improvement offered in MerPO, corroborating our theoretical results. Further, the superior performance of MerPO over existing offline Meta-RL methods suggests that model-based approaches can be more beneficial in offline Meta-RL.

## 2 RELATED WORK

**Offline single-task RL.** Many existing model-free offline RL methods regularize the learnt policy to be close to the behavior policy by, e.g., distributional matching (Fujimoto et al., 2019), support matching (Kumar et al., 2019), importance sampling (Nachum et al., 2019; Liu et al., 2020), learning lower bounds of true Q-values (Kumar et al., 2020). Along a different avenue, model-based algorithms learn policies by leveraging a dynamics model obtained with the offline dataset. (Matsushima et al., 2020) directly constrains the learnt policy to the behavior policy as in model-free algorithms. To penalize the policy for visiting states where the learnt model is likely to be incorrect, MOPO (Yu et al., 2020) and MoREL (Kidambi et al., 2020) modify the learnt dynamics such that the value estimates are conservative when the model uncertainty is above a threshold. To remove the need of uncertainty quantification, COMBO (Yu et al., 2021b) is proposed by combining model-based policy optimization (Janner et al., 2019) and conservative policy evaluation (Kumar et al., 2020).

**Offline Meta-RL.** A few very recent studies have explored the offline Meta-RL. Particularly, (Li et al., 2020a) considers a special scenario where the task identity is spuriously inferred due to biased

datasets, and applies the triplet loss to robustify the task inference with reward relabelling. (Dorfman & Tamar, 2020) extends an online Meta-RL method VariBAD (Zintgraf et al., 2019) to the offline setup, and assumes known reward functions and shared dynamics across tasks. Based on MAML (Finn et al., 2017), (Mitchell et al., 2020) proposes an offline Meta-RL algorithm with advantage weighting loss, and learns initializations for both the value function and the policy, where they consider the offline dataset in the format of full trajectories in order to evaluate the advantage. Based on the off-policy Meta-RL method PEARL (Rakelly et al., 2019), (Li et al., 2020b) combines the idea of deterministic context encoder and behavior regularization, under the assumption of deterministic MDP. Different from the above works, we study a more general offline Meta-RL problem. More importantly, MerPO strikes a right balance between exploring with the meta-policy and exploiting the offline dataset, which guarantees safe performance improvement for new offline tasks.

## 3 PRELIMINARIES

Consider a Markov decision process (MDP) $\mathcal{M} = (\mathcal{S}, \mathcal{A}, T, r, \mu_0, \gamma)$ with state space $\mathcal{S}$, action space $\mathcal{A}$, the environment dynamics $T(s'|s, a)$, reward function $r(s, a)$, initial state distribution $\mu_0$, and $\gamma \in (0, 1)$ is the discount factor. Without loss of generality, we assume that $|r(s, a)| \leq R_{max}$. Given a policy $\pi$, let $d^\pi_{\mathcal{M}}(s) := (1 - \gamma) \sum_{t=0}^\infty \gamma^t P_{\mathcal{M}}(s_t = s|\pi)$ denote the discounted marginal state distribution, where $P_{\mathcal{M}}(s_t = s|\pi)$ denotes the probability of being in state $s$ at time $t$ by rolling out $\pi$ in $\mathcal{M}$. Accordingly, let $d^\pi_{\mathcal{M}}(s, a) := d^\pi_{\mathcal{M}}(s)\pi(a|s)$ denote the discounted marginal state-action distribution, and $J(\mathcal{M}, \pi) := \frac{1}{1-\gamma}\mathbb{E}_{(s,a)\sim d^\pi_{\mathcal{M}}(s,a)}[r(s, a)]$ denote the expected discounted return. The goal of RL is to find the optimal policy that maximizes $J(\mathcal{M}, \pi)$. In offline RL, no interactions with the environment are allowed, and we only have access to a fixed dataset $\mathcal{D} = \{(s, a, r, s')\}$ collected by some unknown behavior policy $\pi_\beta$. Let $d^{\pi_\beta}_{\mathcal{M}}(s)$ be the discounted marginal state distribution of $\pi_\beta$. The dataset $\mathcal{D}$ is indeed sampled from $d^{\pi_\beta}_{\mathcal{M}}(s, a) = d^{\pi_\beta}_{\mathcal{M}}(s)\pi_\beta(a|s)$. Denote $\overline{\mathcal{M}}$ as the empirical MDP induced by $\mathcal{D}$ and $d(s, a)$ as a sample-based version of $d^{\pi_\beta}_{\mathcal{M}}(s, a)$.

In offline Meta-RL, consider a distribution of RL tasks $p(\mathcal{M})$ as in standard Meta-RL (Finn et al., 2017; Rakelly et al., 2019), where each task $\mathcal{M}_n$ is an MDP, i.e., $\mathcal{M}_n = (\mathcal{S}, \mathcal{A}, T_n, r_n, \mu_{0,n}, \gamma)$, with task-shared state and action spaces, and unknown task-specific dynamics and reward function. For each task $\mathcal{M}_n$, no interactions with the environment are allowed and we only have access to an offline dataset $\mathcal{D}_n$, collected by some unknown behavior policy $\pi_{\beta,n}$. The main objective is to learn a meta-policy based on a set of offline training tasks $\{\mathcal{M}_n\}_{n=1}^N$.

**Conservative Offline Model-Based Policy Optimization (COMBO).** Recent model-based offline RL algorithms, e.g., COMBO (Yu et al., 2021b), have demonstrated promising performance on a single offline RL task by combining model-based policy optimization (Janner et al., 2019) and conservative policy evaluation (CQL (Kumar et al., 2020)). Simply put, COMBO first trains a dynamics model $\widehat{T}_\theta(s'|s, a)$ parameterized by $\theta$, via supervised learning on the offline dataset $\mathcal{D}$. The learnt MDP is constructed as $\widehat{\mathcal{M}} := (\mathcal{S}, \mathcal{A}, \widehat{T}, r, \mu_0, \gamma)$. Then, the policy is learnt using $\mathcal{D}$ and model-generated rollouts. Specifically, define the action-value function (Q-function) as $Q^\pi(s, a) := \mathbb{E}\left[\sum_{t=0}^\infty \gamma^t r(s_t, a_t)|s_0 = s, a_0 = a\right]$, and the empirical Bellman operator as: $\widehat{\mathcal{B}}^\pi Q(s, a) = r(s, a) + \gamma \mathbb{E}_{(s,a,s')\sim\mathcal{D}}[Q(s', a')]$, for $a' \sim \pi(\cdot|s')$. To penalize the Q functions in out-of-distribution state-action tuples, COMBO employs conservative policy evaluation based on CQL:

$$\widehat{Q}^{k+1} \leftarrow \arg\min_{Q(s,a)} \beta(\mathbb{E}_{s,a\sim\rho}[Q(s,a)] - \mathbb{E}_{s,a\sim\mathcal{D}}[Q(s,a)]) + \frac{1}{2}\mathbb{E}_{s,a,s'\sim d_f}[(Q(s,a) - \widehat{\mathcal{B}}^\pi\widehat{Q}^k(s,a))^2] \quad (1)$$

where $\rho(s, a) := d^\pi_{\widehat{\mathcal{M}}}(s)\pi(a|s)$ is the discounted marginal distribution when rolling out $\pi$ in $\widehat{\mathcal{M}}$, and $d_f(s, a) = f d^{\pi_\beta}_{\mathcal{M}}(s, a) + (1 - f)\rho(s, a)$ for $f \in [0, 1]$. The Bellman backup $\widehat{\mathcal{B}}^\pi$ over $d_f$ can be interpreted as an $f$-interpolation of the backup operators under the empirical MDP (denoted by $\mathcal{B}^\pi_{\overline{\mathcal{M}}}$) and the learnt MDP (denoted by $\mathcal{B}^\pi_{\widehat{\mathcal{M}}}$). Given the Q-estimation $\widehat{Q}^\pi$, the policy can be learnt by:

$$\pi' \leftarrow \arg\max_\pi \mathbb{E}_{s\sim\rho(s),a\sim\pi(\cdot|s)}[\widehat{Q}^\pi(s, a)]. \quad (2)$$

## 4 MERPO: MODEL-BASED OFFLINE META-RL WITH REGULARIZED POLICY OPTIMIZATION

Learnt dynamics models not only serves as a natural remedy for task structure inference in offline Meta-RL, but also facilitates better exploration of out-of-distribution state-actions by generating

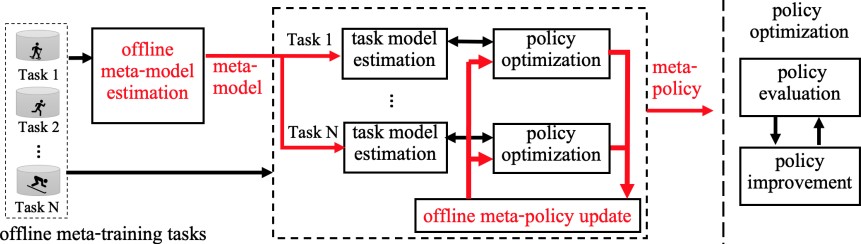

Figure 2: Model-based offline Meta-RL with learning of offline meta-model and offline meta-policy.

synthetic rollouts (Yu et al., 2021b). Thus motivated, we propose a general framework of model-based offline Meta-RL, as depicted in Figure 2. More specifically, the offline meta-model is first learnt by using supervised meta-learning, based on which the task-specific model can be quickly adapted. Then, the main attention of this study is devoted to the learning of an informative meta-policy via bi-level optimization, where 1) a model-based policy optimization approach is leveraged in the inner loop for each task to learn a task-specific policy; and 2) the meta-policy is then updated in the outer loop based on the learnt task-specific policies.

### 4.1 OFFLINE META-MODEL LEARNING

Learning a meta-model based on the set of offline dataset $\{\mathcal{D}_n\}_{n=1}^N$ can be carried out via supervised meta-learning. Many gradient-based meta-learning techniques can be applied here, e.g., MAML (Finn et al., 2017) and Reptile (Nichol et al., 2018). In what follows, we outline the basic idea to leverage the higher-order information of the meta-objective function. Specifically, we consider a proximal meta-learning approach, following the same line as in (Zhou et al., 2019):

$$\min_{\phi_m} \ L_{model}(\phi_m) = \mathbb{E}_{\mathcal{M}_n} \left\{ \min_{\theta_n} \ \left[ \mathbb{E}_{(s,a,s')\sim\mathcal{D}_n}[\log \widehat{T}_{\theta_n}(s'|s,a)] + \eta\|\theta_n - \phi_m\|_2^2 \right] \right\} \quad (3)$$

where the learnt dynamics for each task $\mathcal{M}_n$ is parameterized by $\theta_n$ and the meta-model is parameterized by $\phi_m$. Solving Eq. (3) leads to an offline meta-model.

Given the learnt meta-model $T_{\phi_m^*}$, the dynamics model for an individual offline task $j$ can be found by solving the following problem via gradient descent with initialization $T_{\phi_m^*}$ using $\mathcal{D}_j$, i.e.,

$$\min_{\theta_j} \ \mathbb{E}_{(s,a,s')\sim\mathcal{D}_j}[\log \widehat{T}_{\theta_j}(s'|s,a)] + \eta\|\theta_j - \phi_m^*\|_2^2. \quad (4)$$

Compared to learning the dynamics model from scratch, adapting from $T_{\phi_m^*}$ can quickly generate a dynamics model for task identity inference by leveraging the knowledge from similar tasks, and hence improve the sample efficiency (Finn et al., 2017; Zhou et al., 2019).

### 4.2 OFFLINE META-POLICY LEARNING

In this section, we turn attention to tackle one main challenge in this study: How to learn an informative offline meta-policy in order to achieve the optimal tradeoff between "exploring" the out-of-distribution state-actions by following the meta-policy and "exploiting" the offline dataset by staying close to the behavior policy? Clearly, it is highly desirable for the meta-policy to safely 'explore' out-of-distribution state-action pairs, and for each task to utilize the meta-policy to mitigate the issue of value overestimation.

#### 4.2.1 HOW DO EXISTING PROXIMAL META-RL APPROACHES PERFORM?

Proximal Meta-RL approaches have demonstrated remarkable performance in the online setting (e.g., (Wang et al., 2020)), by explicitly regularizing the task-specific policy close to the meta-policy. We first consider the approach that applies the online Proximal Meta-RL method directly to devise offline Meta-RL, which would lead to:

$$\max_{\pi_c} \ \mathbb{E}_{\mathcal{M}_n} \left\{ \max_{\pi_n} \ \left[ \mathbb{E}_{\substack{s\sim\rho_n, \\ a\sim\pi_n(\cdot|s)}} \left[ \hat{Q}_n^\pi(s,a) \right] - \lambda D(\pi_n, \pi_c) \right] \right\} \quad (5)$$

where $\pi_c$ is the offline meta-policy, $\pi_n$ is the task-specific policy, $\rho_n$ is the state marginal of $\rho_n(s,a)$ for task $n$ and $D(\cdot,\cdot)$ is some distance measure between two probability distributions. To alleviate value overestimation, conservative policy evaluation can be applied to learn $\hat{Q}_n^\pi$ by using Eq. (1).

Intuitively, Eq. (5) corresponds to generalizing COMBO to the multi-task setting, where a meta policy $\pi_c$ is learned to regularize the within-task policy optimization.

To get a sense of how the meta-policy learnt using Eq. (5) performs, we evaluate its performance in an offline variant of standard Meta-RL benchmark Walker-2D-Params with good-quality datasets, and evaluate the testing performance of the task-specific policy after fine-tuning based on the learnt meta-policy, with respect to the meta-training steps. As can be seen in Figure 3, the proximal Meta-RL algorithm Eq. (5) performs surprisingly poorly and fails to learn an informative meta-policy, despite conservative policy evaluation being applied in within-task policy optimization to deal with the value overestimation. In particular, the testing performance degrades along with the meta-training process, implying that the quality of the learnt meta-policy is in fact decreasing.

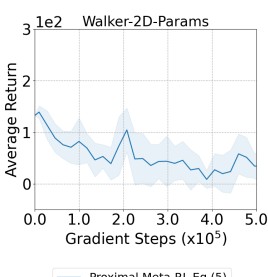

Figure 3: Performance of proximal Meta-RL Eq. (5).

***Why does the proximal Meta-RL method in Eq. (5) perform poorly in offline Meta-RL, even with conservative policy evaluation?*** To answer this, it is worth to take a closer look at the within-task policy optimization in Eq. (5), which is given as follows:

$$\pi_n \leftarrow \arg\max_{\pi_n} \mathbb{E}_{s \sim \rho_n, a \sim \pi_n(\cdot|s)}[\widehat{Q}_n^\pi(s,a)] - \lambda D(\pi_n, \pi_c). \qquad (6)$$

Clearly, the performance of Eq. (6) depends heavily on the quality of the meta-policy $\pi_c$. A poor meta-policy may have negative impact on the performance and result in a task-specific policy $\pi_n$ that is even outperformed by the behaviour policy $\pi_{\beta,n}$. Without online exploration, the quality of $\pi_n$ could not be improved, which in turn leads to a worse meta-policy $\pi_c$ through Eq. (5). The iterative meta-training process would eventually result in the performance degradation in Figure 3.

In a nutshell, simply following the meta-policy may lead to worse performance of offline tasks when $\pi_\beta$ is a better policy than $\pi_c$. Since it is infeasible to guarantee the superiority of the meta-policy *a priori*, it is necessary to balance the tradeoff between exploring with the meta-policy and exploiting the offline dataset, in order to guarantee the performance improvement of new offline tasks.

### 4.2.2 SAFE POLICY IMPROVEMENT WITH META-REGULARIZATION

To tackle the above challenge, we next devise a novel regularized policy improvement for within-task policy optimization of task $n$, through a weighted interpolation of two different regularizers based on the behavior policy $\pi_{\beta,n}$ and the meta-policy $\pi_c$, given as follows:

$$\pi_n \leftarrow \arg\max_{\pi_n} \mathbb{E}_{s \sim \rho_n, a \sim \pi_n(\cdot|s)}[\widehat{Q}_n^\pi(s,a)] - \lambda\alpha D(\pi_n, \pi_{\beta,n}) - \lambda(1-\alpha)D(\pi_n, \pi_c), \qquad (7)$$

for some $\alpha \in [0,1]$. Here, $\alpha$ controls the trade-off between staying close to the behavior policy and following the meta-policy to "explore" out-of-distribution state-actions. Intuitively, as $\alpha$ is closer to 0, the policy improvement is less conservative and tends to improve the task-specific policy $\pi_n$ towards the actions in $\pi_c$ that have highest estimated Q-values. Compared to Eq. (6), the exploration penalty induced by $D(\pi_n, \pi_{\beta,n})$ serves as a safeguard and stops $\pi_n$ following $\pi_c$ over-optimistically.

**Safe Policy Improvement Guarantee.** Based on conservative policy evaluation Eq. (1) and regularized policy improvement Eq. (7), we have the meta-regularized model-based actor-critic method (RAC), as outlined in Algorithm 1. Note that different distribution distance measures can be used in Eq. (7). In this work, we theoretically show that the policy $\pi_n(a|s)$ learnt by RAC is a safe improvement over both the behavior policy $\pi_{\beta,n}$ and the meta-policy $\pi_c$ on the underlying MDP $\mathcal{M}_n$, when using the maximum total-variation distance for $D(\pi_1, \pi_2)$, i.e., $D(\pi_1, \pi_2) := \max_s D_{TV}(\pi_1 \| \pi_2)$.

---

**Algorithm 1** RAC

1: Train dynamics model $\widehat{T}_{\theta_n}$ using $\mathcal{D}_n$;
2: **for** $k = 1, 2, ...$ **do**
3:     Perform model rollouts starting from states in $\mathcal{D}_n$ and add into $\mathcal{D}_{model,n}$;
4:     Policy evaluation by recursively solving Eq. (1) using $\mathcal{D}_n \cup \mathcal{D}_{model,n}$;
5:     Improve policy by solving Eq. (7);
6: **end for**

---

For convenience, define $\nu_n(\rho, f) = \mathbb{E}_\rho\left[(\rho(s,a) - d_n(s,a))/d_{f,n}(s,a)\right]$, and let $\delta \in (0, 1/2)$. We have the following important result on the safe policy improvement achieved by $\pi_n(a|s)$.

**Theorem 1.** *(a) Let $\epsilon = \frac{\beta[\nu_n(\rho^{\pi_n}, f) - \nu_n(\rho^{\pi_{\beta,n}}, f)]}{2\lambda(1-\gamma)D(\pi_n, \pi_{\beta,n})}$. If $\nu_n(\rho^{\pi_n}, f) - \nu_n(\rho^{\pi_{\beta,n}}, f) > 0$ and $\alpha \in \left(\max\{\frac{1}{2} - \epsilon, 0\}, \frac{1}{2}\right)$, then $J(\mathcal{M}_n, \pi_n) \geq \max\{J(\mathcal{M}_n, \pi_c) + \xi_1, J(\mathcal{M}_n, \pi_{\beta,n}) + \xi_2\}$ holds with probability at least $1 - 2\delta$, where both $\xi_1$ and $\xi_2$ are positive for large enough $\beta$ and $\lambda$;*

*(b) More generally, we have that $J(\mathcal{M}_n, \pi_n) \geq \max\{J(\mathcal{M}_n, \pi_c) + \xi_1, J(\mathcal{M}_n, \pi_{\beta,n}) + \xi_2\}$ holds with probability at least $1 - 2\delta$, when $\alpha \in (0, 1/2)$, where $\xi_1$ is positive for large enough $\lambda$.*

**Remark 1.** The expressions of $\xi_1$ and $\xi_2$ are involved and can be found in Eq. (14) and Eq. (15) in the appendix. In part (a) of Theorem 1, both $\xi_1$ and $\xi_2$ are positive for large enough $\beta$ and $\lambda$, pointing to guaranteed improvements over $\pi_c$ and $\pi_{\beta,n}$. Due to the fact that the dynamics $T_{\widehat{\mathcal{M}_n}}$ learnt via supervised learning is close to the true dynamics $T_{\mathcal{M}_n}$ on the states visited by the behavior policy $\pi_{\beta,n}$, $d_{\widehat{\mathcal{M}_n}}^{\pi_{\beta,n}}(s, a)$ is close to $d_{\mathcal{M}_n}^{\pi_{\beta,n}}(s, a)$ and $\rho^{\pi_{\beta,n}}$ is close to $d_n(s, a)$, indicating that the condition $\nu_n(\rho^{\pi_n}, f) - \nu_n(\rho^{\pi_{\beta,n}}, f) > 0$ is expected to hold in practical scenarios (Yu et al., 2021b). For more general cases, a slightly weaker result can be obtained in part (b) of Theorem 1, where $\xi_1$ is positive for large enough $\lambda$ and $\xi_2$ can be negative.

**Remark 2.** Intuitively, the selection of $\alpha$ balances the impact of $\pi_{\beta,n}$ and $\pi_c$, while delicately leaning toward the meta-policy $\pi_c$ because $\pi_{\beta,n}$ has played an important role in policy evaluation to find a lower bound of Q-value. As a result, *Eq. (7) maximizes the true Q-value while implicitly regularized by a weighted combination, instead of $\alpha$-interpolation, between $D(\pi_n, \pi_{\beta,n})$ and $D(\pi_n, \pi_c)$, where the weights are carefully balanced through $\alpha$.* In particular, in the tabular setting, the conservative policy evaluation in Eq. (1) corresponds to penalizing the Q estimation (Yu et al., 2021b):

$$\widehat{Q}_n^{k+1}(s, a) = \widehat{\mathcal{B}}^\pi \widehat{Q}_n^k(s, a) - \frac{\beta[\rho(s, a) - d_n(s, a)]}{d_{f,n}(s, a)}. \tag{8}$$

Clearly, $\epsilon$ increases with the value of the penalty term in Eq. (8). As a result, when the policy evaluation Eq. (1) is overly conservative, the lower bound of $\alpha$ will be close to 0, and hence the regularizer based on the meta-policy $\pi_c$ can play a bigger role so as to encourage the "exploration" of out-of-distribution state-actions following the guidance of $\pi_c$. On the other hand, when the policy evaluation Eq. (1) is less conservative, the lower bound of $\alpha$ will be close to $\frac{1}{2}$, and the regularizer based on $\pi_{\beta,n}$ will have more impact, leaning towards "exploiting" the offline dataset. In fact, the introduction of 1) behavior policy-based regularizer and 2) the interpolation for modeling the interaction between the behavior policy and the meta-policy, is the key to prove Theorem 1.

**Practical Implementation.** In practice, we can use the KL divergence to replace the total variation distance between policies, based on Pinsker's Inequality: $\|\pi_1 - \pi_2\| \leq \sqrt{2D_{KL}(\pi_1 \| \pi_2)}$. Moreover, since the behavior policy $\pi_{\beta,n}$ is typically unknown, we can use the reverse KL-divergence between $\pi_n$ and $\pi_{\beta,n}$ to circumvent the estimation of $\pi_{\beta,n}$, following the same line as in (Fakoor et al., 2021):

$$D_{KL}(\pi_{\beta,n} \| \pi_n) = \mathbb{E}_{a \sim \pi_{\beta,n}}[\log \pi_{\beta,n}(a|s)] - \mathbb{E}_{a \sim \pi_{\beta,n}}[\log \pi_n(a|s)]$$

$$\propto -\mathbb{E}_{a \sim \pi_{\beta,n}}[\log \pi_n(a|s)] \approx -\mathbb{E}_{(s,a) \sim \mathcal{D}_n}[\log \pi_n(a|s)].$$

Then, the task-specific policy can be learnt by solving the following problem:

$$\max_{\pi_n} \mathbb{E}_{s \sim \rho_n, a \sim \pi_n(\cdot|s)} \left[\widehat{Q}_n^\pi(s, a)\right] + \lambda\alpha \mathbb{E}_{(s,a) \sim \mathcal{D}_n}[\log \pi_n(a|s)] - \lambda(1 - \alpha)D_{KL}(\pi_n \| \pi_c). \tag{9}$$

### 4.2.3 OFFLINE META-POLICY UPDATE

Built on RAC, the offline meta-policy $\pi_c$ is updated by taking the following two steps, in an iterative manner: 1) (*inner loop*) given the meta-policy $\pi_c$, RAC is run for each training task to obtain the task-specific policy $\pi_n$; 2) (*outer loop*) based on $\{\pi_n\}_n$, $\pi_c$ is updated by solving:

$$\max_{\pi_c} \mathbb{E}_{\mathcal{M}_n} \left\{ \mathbb{E}_{\substack{s \sim \rho_n, \\ a \sim \pi_n(\cdot|s)}} \left[\widehat{Q}_n^\pi(s, a)\right] + \lambda\alpha \mathbb{E}_{(s,a) \sim \mathcal{D}_n}[\log \pi_n(a|s)] - \lambda(1 - \alpha)D_{KL}(\pi_n \| \pi_c) \right\} \tag{10}$$

where both $\rho_n$ and $\widehat{Q}_n^\pi$ are from the last iteration of the inner loop for each training task. By using RAC in the inner loop for within-task policy optimization, the learnt task-specific policy $\pi_n$ and the meta-policy $\pi_c$ work in concert to regularize the policy search for each other, and improve akin to 'positive feedback'. Here the regularizer based on the behavior policy serves an important initial force to boost the policy optimization against the ground: RAC in the inner loop aims to improve the task-specific policy over the behavior policy at the outset and the improved task-specific policy consequently regularizes the meta-policy search as in Eq. (10), leading to a better meta-policy eventually. Noted that a meta-Q network is learnt using first-order meta-learning to initialize task-specific Q networks. It is worth noting that different tasks can have different values of $\alpha$ to capture the heterogeneity of dataset qualities across tasks.

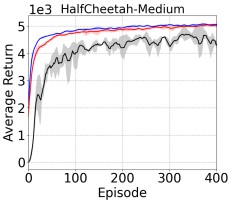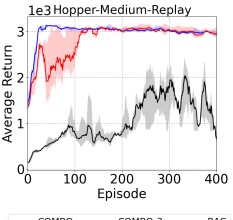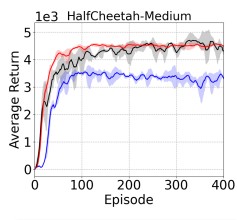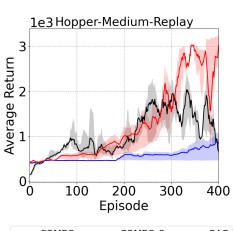

Figure 4: Performance comparison among COMBO, COMBO-3 and RAC, with good-quality meta-policy (two figures on the left) and poor-quality meta-policy (two figures on the right).

In a nutshell, the proposed model-based offline Meta-RL with regularized Policy Optimization (MerPO) is built on two key steps: 1) learning the offline meta-model via Eq. (3) and 2) learning the offline meta-policy via Eq. (10). The details are presented in Algorithm 2 in the appendix.

### 4.3 MERPO-BASED POLICY OPTIMIZATION FOR NEW OFFLINE RL TASK

Let $T_{\phi_m^*}$ and $\pi_c^*$ be the offline meta-model and the offline meta-policy learnt by MerPO. For a new offline RL task, the task model can be quickly adapted based on Eq. (4), and the task-specific policy can be obtained based on $\pi_c^*$ using the within-task policy optimization module RAC. Appealing to Theorem 1, we have the following result on MerPO-based policy learning on a new task.

**Proposition 1.** *Consider a new offline RL task with the true MDP $\mathcal{M}$. Suppose $\pi_o$ is the MerPO-based task-specific policy, learnt by running RAC over the meta-policy $\pi_c^*$. If $\epsilon = \frac{\beta[\nu(\rho^{\pi_o}, f) - \nu(\rho^{\pi_\beta}, f)]}{2\lambda(1-\gamma)D(\pi_o, \pi_\beta)} \geq 0$ and $\alpha \in \left(\max\{\frac{1}{2} - \epsilon, 0\}, \frac{1}{2}\right)$, then $\pi_o$ achieves the safe performance improvement over both $\pi_c^*$ and $\pi_\beta$, i.e., $J(\mathcal{M}, \pi_o) > \max\{J(\mathcal{M}, \pi_c^*), J(\mathcal{M}, \pi_\beta)\}$ holds with probability at least $1 - 2\delta$, for large enough $\beta$ and $\lambda$.*

Proposition 1 indicates that MerPO-based policy optimization for learning task-specific policy guarantees a policy with higher rewards than both the behavior policy and the meta-policy. This is particularly useful in the following two scenarios: 1) the offline dataset is collected by some poor behavior policy, but the meta-policy is a good policy; and 2) the meta-policy is inferior to a good behavior policy.

## 5 EXPERIMENTS

In what follows, we first evaluate the performance of RAC for within-task policy optimization on an offline RL task to validate the safe policy improvement, and then examine how MerPO performs when compared to state-of-art offline Meta-RL algorithms. Due to the space limit, we relegate additional experiments to the appendix.

### 5.1 PERFORMANCE EVALUATION OF RAC

**Setup.** We evaluate RAC on several continuous control tasks in the D4RL benchmark (Fu et al., 2020) from the Open AI Gym (Brockman et al., 2016), and compare its performance to 1) COMBO (where no meta-policy is leveraged) and 2) COMBO with policy improvement Eq. (6) (namely, COMBO-3), under different qualities of offline datasets and different qualities of meta-policy (good and poor). For illustrative purpose, we use a random policy as a poor-quality meta-policy, and choose the learnt policy after 200 episodes as a better-quality meta-policy. We evaluate the average return over 4 random seeds after each episode with 1000 gradient steps.

**Results.** As shown in Figure 4, RAC can achieve comparable performance with COMBO-3 given a good-quality meta-policy, and both clearly outperform COMBO. Besides, the training procedure is also more stable and converges more quickly as expected when regularized with the meta-policy. When regularized by a poor-quality meta-policy, that is significantly worse than the behavior policy in all environments, the performance of COMBO-3 degrades dramatically. However, RAC outperforms COMBO even when the meta-policy is a random policy. In a nutshell, RAC consistently

achieves the best performance in various setups and demonstrates compelling robustness against the quality of the meta-policy, for suitable parameter selections ($\alpha = 0.4$ in Figure 4).

**Impact of $\alpha$.** As shown in Theorem 1, the selection of $\alpha$ is important to guarantee the safe policy improvement property of RAC. Therefore, we next examine the impact of $\alpha$ on the performance of RAC under different qualities of datasets and meta-policy. More specifically, we consider four choices of $\alpha$: $\alpha = 0, 0.4, 0.7, 1$. Here, $\alpha = 0$ corresponds to COMBO-3, i.e., regularized by the meta-policy only, and the policy improvement step is regularized by the behavior policy only when

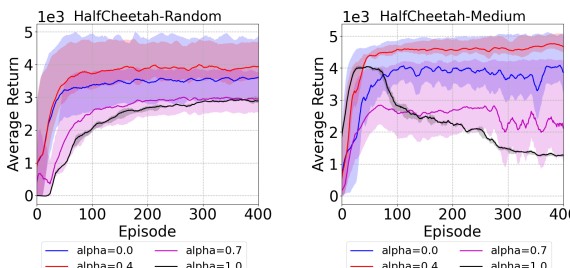

Figure 5: Impact of $\alpha$ on the performance of RAC under different qualities of offline datasets.

$\alpha = 1$. Figure 5 shows the average return of RAC over different qualities of meta-policies under different qualities of the offline datasets. It is clear that RAC achieves the best performance when $\alpha = 0.4$ among the four selections of $\alpha$, corroborating the result in Theorem 1. In general, the performance of RAC is stable for $\alpha \in [0.3, 0.5]$ in our experiments.

## 5.2 PERFORMANCE EVALUATION OF MERPO

**Setup.** To evaluate the performance of MerPO, we follow the setups in the literature (Rakelly et al., 2019; Li et al., 2020b) and consider continuous control meta-environments of robotic locomotion. More specifically, tasks has different transition dynamics in Walker-2D-Params and Point-Robot-Wind, and different reward functions in Half-Cheetah-Fwd-Back and Ant-Fwd-Back. We collect the offline dataset for each task by following the same line as in (Li et al., 2020b). We consider the following baselines: (1) FOCAL (Li et al., 2020b), a model-free offline Meta-RL approach based on a deterministic context encoder that achieves the state-of-the-art performance; (2) MBML (Li et al., 2020a), an offline multi-task RL approach with metric learning; (3) Batch PEARL, which modifies PEARL (Rakelly et al., 2019) to train and test from offline datasets without exploration; (4) Contextual BCQ (CBCQ), which is a task-augmented variant of the offline RL algorithm BCQ (Fujimoto et al., 2019) by integrating a task latent variable into the state information. We train on a set of offline RL tasks, and evaluate the performance of the learnt meta-policy during the training process on a set of unseen testing offline RL tasks.

**Fixed $\alpha$ vs Adaptive $\alpha$.** We consider two implementations of MerPO based on the selection of $\alpha$. 1) MerPO: $\alpha$ is fixed as 0.4 for all tasks; 2) MerPO-Adp: at each iteration $k$, given the task-policy $\pi_n^k$ for task $n$ and the meta-policy $\pi_c^k$ at iteration $k$, we update $\alpha_n^k$ using one-step gradient descent to minimize the following problem.

$$\min_{\alpha_n^k} (1 - \alpha_n^k)(D(\pi_n^k, \pi_{\beta, n}) - D(\pi_n^k, \pi_c^k)), \text{ s.t. } \alpha_n^k \in [0.1, 0.5]. \tag{11}$$

The idea is to adapt $\alpha_n^k$ in order to balance between $D(\pi_n^k, \pi_{\beta, n})$ and $D(\pi_n^k, \pi_c^k)$, because Theorem 1 implies that the safe policy improvement can be achieved when the impacts of the meta-policy and the behavior policy are well balanced. Specifically, at iteration $k$ for each task $n$, $\alpha_n^k$ is increased when the task-policy $\pi_n^k$ is closer to the meta-policy $\pi_c^k$, and is decreased when $\pi_n^k$ is closer to the behavior policy. Note that $\alpha_n^k$ is constrained in the range $[0.1, 0.5]$ as suggested by Theorem 1.

**Results.** As illustrated in Figure 6, MerPO-Adp yields the best performance, and both MerPO-Adp and MerPO achieve better or comparable performance in contrast to existing offline Meta-RL approaches. Since the meta-policy changes during the learning process and the qualities of the behavior policies vary across different tasks, MerPO-Adp adapts $\alpha$ across different iterations and tasks so as to achieve a 'local' balance between the impacts of the meta-policy and the behavior policy. As expected, MerPO-Adp can perform better than MerPO with a fixed $\alpha$. Here the best testing performance for the baseline algorithms is selected over different qualities of offline datasets.

**Ablation Study.** We next provide ablation studies by answering the following questions.

**(1) Is RAC important for within-task policy optimization?** To answer this question, we compare MerPO with the approach Eq. (5) where the within-task policy optimization is only regularized by

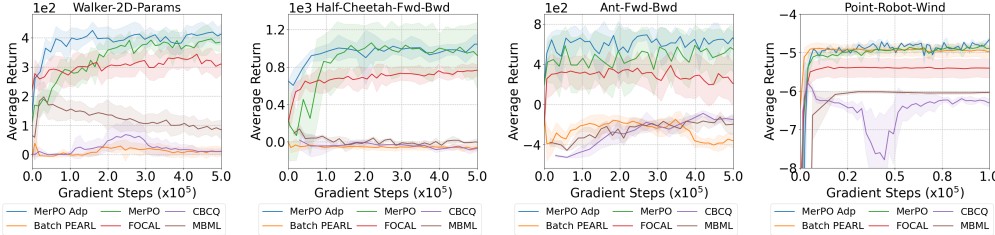

Figure 6: Performance comparison in terms of the average return in different environments. Clearly, MerPO Adp and MerPO achieve better or comparable performance than the baselines.

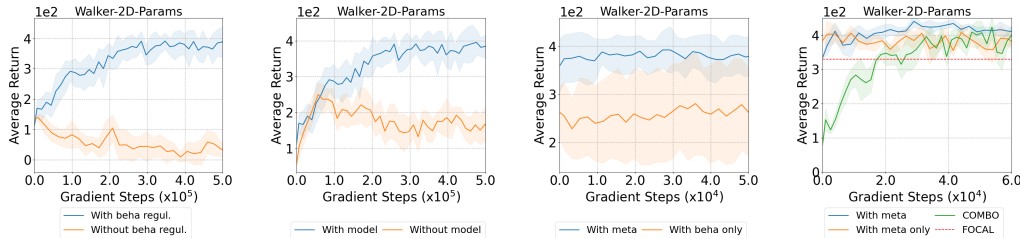

(a) Impact of RAC module. (b) Impact of model utilization. (c) Performance under different data qualities. (d) Testing performance for expert dataset.

Figure 7: Ablation study of MerPO in Walker-2D-Params.

the meta-policy. As shown in Figure 7(a), with the regularization based on the behavior policy in RAC, MerPO performs significantly better than Eq. (5), implying that the safe policy improvement property of RAC enables MerPO to continuously improve the meta-policy.

**(2) Is learning the dynamics model important?** Without the utilization of models, the within-task policy optimization degenerates to CQL (Kumar et al., 2020) and the Meta-RL algorithm becomes a model-free approach. Figure 7(b) shows the performance comparison between the cases whether the dynamics model is utilized. It can be seen that the performance without model utilization is much worse than that of MerPO. This indeed makes sense because the task identity inference (Dorfman & Tamar, 2020; Li et al., 2020a;b) is a critical problem in Meta-RL. Such a result also aligns well with a recently emerging trend in supervised meta-learning to improve the generalization ability by augmenting the tasks with "more data" (Rajendran et al., 2020; Yao et al., 2021).

**(3) How does MerPO perform in unseen offline tasks under different data qualities?** We evaluate the average return in unseen offline tasks with different data qualities, and compare the performance between (1) MerPO with $\alpha = 0.4$ ("With meta") and (2) Run a variant of COMBO with behavior-regularized policy improvement, i.e., $\alpha = 1$ ("With beha only"). For a fair comparison, we initialize the policy network with the meta-policy in both cases. As shown in Figure 7(c), the average performance of "With meta" over different data qualities is much better than that of "With beha only". More importantly, for a new task with expert data, MerPO ("With meta") clearly outperforms COMBO as illustrated in Figure 7(d), whereas the performance of FOCAL is worse than COMBO.

## 6 CONCLUSION

In this work, we study offline Meta-RL aiming to strike a right balance between "exploring" the out-of-distribution state-actions by following the meta-policy and "exploiting" the offline dataset by staying close to the behavior policy. To this end, we propose a model-based offline Meta-RL approach, namely MerPO, which learns a meta-model to enable efficient task model learning and a meta-policy to facilitate safe exploration of out-of-distribution state-actions. Particularly, we devise RAC, a meta-regularized model-based actor-critic method for within-task policy optimization, by using a weighted interpolation between two regularizers, one based on the behavior policy and the other on the meta-policy. We theoretically show that the learnt task-policy via MerPO offers safe policy improvement over both the behavior policy and the meta-policy. Compared to existing offline Meta-RL methods, MerPO demonstrates superior performance on several benchmarks, which suggests a more prominent role of model-based approaches in offline Meta-RL.

## ACKNOWLEDGEMENT

This work is supported in part by NSF Grants CNS-2003081, CNS-2203239, CPS-1739344, and CCSS-2121222.

## REPRODUCIBILITY STATEMENT

For the theoretical results presented in the main text, we state the full set of assumptions of all theoretical results in Appendix B, and include the complete proofs of all theoretical results in Appendix C. For the experimental results presented in the main text, we include the code in the supplemental material, and specify all the training details in Appendix A. For the datasets used in the main text, we also give a clear explanation in Appendix A.

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

Table 1: Hyperparameters for RAC.

| Hyperparameters | Halfcheetah | Hopper | Walker2d |
|---|---|---|---|
| Discount factor | 0.99 | 0.99 | 0.99 |
| Sample batch size | 256 | 256 | 256 |
| Real data ratio | 0.5 | 0.5 | 0.5 |
| Model rollout length | 5 | 5 | 1 |
| Critic lr | 3e-4 | 3e-4 | 1e-4 |
| Actor lr | 1e-4 | 1e-4 | 1e-5 |
| Model lr | 1e-3 | 1e-3 | 1e-3 |
| Optimizer | Adam | Adam | Adam |
| $\beta$ | 1 | 1 | 10 |
| Max entropy | True | True | True |
| $\lambda$ | 1 | 1 | 1 |

## A  EXPERIMENTAL DETAILS

### A.1  META ENVIRONMENT DESCRIPTION

- Walker-2D-Params: Train an agent to move forward. Different tasks correspond to different randomized dynamcis parameters.

- Half-Cheeta-Fwd-Back: Train a Cheetah robot to move forward or backward, and the reward function depends on the moving direction. All tasks have the same dynamics model but different reward functions.

- Ant-Fwd-Back: Train an Ant robot to move forward or backward, and the reward function depends on the moving direction. All tasks have the same dynamics model but different reward functions.

- Poing-Robot-Wind: Point-Robot-Wind is a variant of Sparse-Point-Robot (Li et al., 2020b), a 2D navigation problem introduced in (Rakelly et al., 2019), where each task is to guide a point robot to navigate to a specific goal location on the edge of a semi-circle from the origin. In Point-Robot-Wind, each task is affected by a distinct "wind" uniformly sampled from $[-0.05, 0.05]^2$, and hence differs in the transition dynamics.

### A.2  IMPLEMENTATION DETAILS AND MORE EXPERIMENTS

#### A.2.1  EVALUATION OF RAC

**Model learning.** Following the same line as in (Janner et al., 2019; Yu et al., 2020; 2021b), the dynamics model for each task is represented as a probabilistic neural network that takes the current state-action as input and outputs a Gaussian distribution over the next state and reward:
$$\widehat{T}_\theta(s_{t+1}, r | s, a) = \mathcal{N}(\mu_\theta(s_t, a_t), \Sigma_\theta(s_t, a_t)).$$
An ensemble of 7 models is trained independently using maximum likelihood estimation, and the best 5 models are picked based on the validation prediction error using a held-our set of the offline dataset. Each model is represented by a 4-layer feedforward neural network with 256 hidden units. And one model will be randomly selected from the best 5 models for model rollout.

**Policy optimization.** We represent both Q-network and policy network as a 4-layer feedforward neural network with 256 hidden units, and use clipped double Q-learning (Fujimoto et al., 2018) for Q backup update. A max entropy term is also included to the value function for computing the target Q value as in SAC (Haarnoja et al., 2018). The hyperparameters used for evaluating the performance of RAC are described in Table 1.

**Additional experiments.** We also evaluate the performance of RAC in Walker2d under different qualities of the meta-policy. As shown in Figure 8(a) and 8(b), RAC achieves the best performance in both scenarios, compared to COMBO and COMBO-3. Particularly, the performance of COMBO-3 in Figure 8(a) degrades in the later stage of training because the meta-policy is not superior over the behavior policy in this case. In stark contrast, the performance of RAC is consistently better, as it provides a safe policy improvement guarantee over both the behavior policy and the meta-policy.

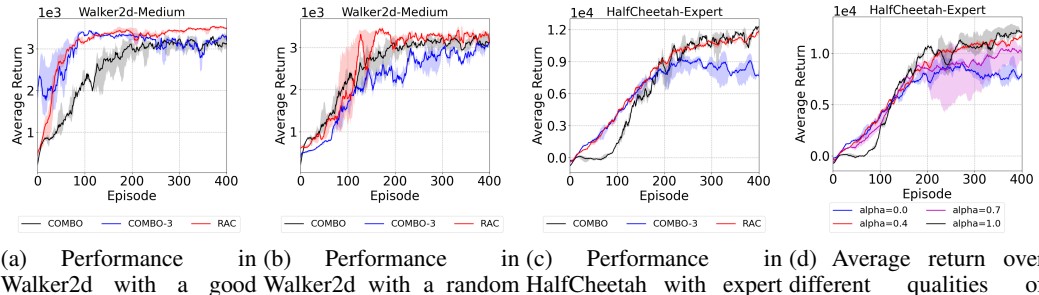

(a) Performance in Walker2d with a good meta-policy.

(b) Performance in Walker2d with a random meta-policy.

(c) Performance in HalfCheetah with expert offline dataset.

(d) Average return over different qualities of meta-policies under expert dataset for different choices of $\alpha$.

Figure 8: Performance evaluation of RAC.

Beside, we also compare the performance of these three algorithms under an expert behavior policy in Figure 8(c), where a meta-policy usually interferes the policy optimization and drags the learnt policy away from the expert policy. As expected, RAC can still achieve comparable performance with COMBO, as a result of safe policy improvement over the behavior policy for suitable parameter selections.

We examine the impact of $\alpha$ on the performance of RAC under different qualities of the meta-policy for HalfCheetah with expert data. In this case, the meta-policy is a worse policy compared to the behavior policy. As shown in Figure 8(d), the performance $\alpha = 0.4$ is comparable to the case of $\alpha = 1$ where the policy improvement step is only regularized based on the behavior policy, and clearly better than the other two cases.

### A.2.2 EVALUATION OF MERPO

**Data collection.** We collect the offline dataset for each task by training a stochastic policy network using SAC (Haarnoja et al., 2018) for that task and rolling out the policies saved at each checkpoint to collect trajectories. Different checkpoints correspond to different qualities of the offline datasets. When training with MerPO, we break the trajectories into independent tuples $\{s_i, a_i, r_i, s_i'\}$ and store in a replay buffer. Therefore, the offline dataset for each task does not contain full trajectories over entire episodes, but merely individual transitions collected by some behavior policy.

**Setup.** For each testing task, we obtain the task-specific policy through quick adaptation using the within-task policy optimization method RAC, based on its own offline dataset and the learnt meta-policy, and evaluate the average return of the adapted policy over 4 random seeds. As shown earlier, we take $\alpha = 0.4$ for all experiments about MerPO. In MerPO-Adp, we initialize $\alpha$ with 0.4 and update with a learning rate of $1e - 4$.

**Meta-model learning.** Similar as in section A.2.1, for each task we quickly adapt from the meta-model to obtain an ensemble of 7 models and pick the best 5 models based on the validation error. The neural network used for representing the dynamics model is same with that in section A.2.1.

**Meta-policy learning.** As in RAC, we represent the task Q-network, the task policy network and the meta-policy network as a 5-layer feedforward neural network with 300 hidden units, and use clipped double Q-learning (Fujimoto et al., 2018) for within task Q backup update. For each task, we also use dual gradient descent to automatically tune both the parameter $\beta$ for conservative policy evaluation and the parameter $\lambda$ for regularized policy improvement:

- Tune $\beta$. Before optimizing the Q-network in policy evaluation, we first optimize $\beta$ by solving the following problem:

$$\min_Q \max_{\beta \geq 0} \beta(\mathbb{E}_{s,a\sim\rho}[Q(s,a)] - \mathbb{E}_{s,a\sim\mathcal{D}}[Q(s,a)] - \tau) + \frac{1}{2}\mathbb{E}_{s,a,s'\sim d_f}[(Q(s,a) - \widehat{\mathcal{B}}^\pi \widehat{Q}^k(s,a))^2].$$

  Intuitively, the value of $\beta$ will be increased to penalty the Q-values for out-of-distribution state-actions if the difference $\mathbb{E}_{s,a\sim\rho}[Q(s,a)] - \mathbb{E}_{s,a\sim\mathcal{D}}[Q(s,a)]$ is larger than some threshold value $\tau$.

Table 2: Hyperparameters for MerPO.

| Hyperparameters | Walker-2D-Params | Half-Cheetah-Fwd-Back | Ant-Fwd-Back | Point-Robot-Wind |
|---|---|---|---|---|
| Discount factor | 0.99 | 0.99 | 0.99 | 0.9 |
| Sample batch size | 256 | 256 | 256 | 256 |
| Task batch size | 8 | 2 | 2 | 8 |
| Real data ratio | 0.5 | 0.5 | 0.5 | 0.5 |
| Model rollout length | 1 | 1 | 1 | 1 |
| Inner critic lr | 1e-3 | 1e-3 | 8e-4 | 1e-3 |
| Inner actor lr | 1e-3 | 5e-4 | 5e-4 | 1e-3 |
| Inner steps | 10 | 10 | 10 | 10 |
| Outer critic lr | 1e-3 | 1e-3 | 1e-3 | 1e-3 |
| Outer actor lr | 1e-3 | 1e-3 | 1e-3 | 1e-3 |
| Meta-q lr | 1e-3 | 1e-3 | 1e-3 | 1e-3 |
| Task model lr | 1e-4 | 1e-4 | 1e-4 | 1e-4 |
| Meta-model lr | 5e-2 | 5e-2 | 5e-2 | 5e-2 |
| Model adaptation steps | 25 | 25 | 25 | 25 |
| Optimizer | Adam | Adam | Adam | Adam |
| Auto-tune $\lambda$ | True | True | True | True |
| $\lambda$ lr | 1 | 1 | 1 | 1 |
| $\lambda$ initial | 5 | 100 | 100 | 5 |
| Target divergence | 0.05 | 0.05 | 0.05 | 0.05 |
| Auto-tune $\beta$ | True | True | True | True |
| $\log \beta$ lr | 1e-3 | 1e-3 | 1e-3 | 1e-3 |
| $\log \beta$ initial | 0 | 0 | 0 | 0 |
| Q difference threshold | 5 | 10 | 10 | 10 |
| Max entropy | True | True | True | True |
| $\alpha$ | 0.4 | 0.4 | 0.4 | 0.4 |
| Testing adaptation steps | 100 | 100 | 100 | 100 |
| # training tasks | 20 | 2 | 2 | 40 |
| # testing tasks | 5 | 2 | 2 | 10 |

- Tune $\lambda$. Similarly, we can optimize $\lambda$ in policy improvement by solving the following problem:

$$\max_{\pi} \min_{\lambda \geq 0} \mathbb{E}_{s \sim \rho(s), a \sim \pi(\cdot|s)}[\widehat{Q}^{\pi}(s,a)] - \lambda \alpha [D(\pi, \pi_\beta) - D_{target}] - \lambda(1-\alpha)[D(\pi, \pi_c) - D_{target}].$$

Intuitively, the value of $\lambda$ will be increased so as to have stronger regularizations if the divergence is larger than some threshold $D_{target}$, and decreased if the divergence is smaller than $D_{target}$.

Besides, we also build a meta-Q network $Q_{meta}$ over the training process as an initialization of the task Q networks to facilitate the within task policy optimization. At the $k$-th meta-iteration for meta-policy update, the meta-Q network is also updated using the average Q-values of current batch $B$ of training tasks with meta-q learning rate $\xi_q$, i.e.,

$$Q_{meta}^{k+1} = Q_{meta}^k - \xi_q [Q_{meta}^k - \frac{1}{|B|} \sum_{n \in B} Q_n].$$

Therefore, we initialize the task Q networks and the task policy with the meta-Q network and the meta-policy, respectively, for within task policy optimization during both meta-training and meta-testing. The hyperparameters used in evaluation of MerPO are listed in Table 2.

For evaluating the performance improvement in a single new offline task, we use a smaller learning rate of $8e-5$ for the Q network and the policy network update.

### A.2.3 MORE EXPERIMENTS.

We also evaluate the impact of the utilization extent of the learnt model, by comparing the performance of MerPO under different cases of real data ratio, i.e., the ratio of the data from the offline dataset in the data batch for training. As shown in Figure 9(a), the performance of MerPO can be further boosted with a more conservative utilization of the model.

To understand how much benefit MerPO can bring for policy learning in unseen offline RL tasks, we compare the performance of the following cases with respect to the gradient steps taken for learning in unseen offline RL tasks: (1) Initialize the task policy network with the meta-policy and run RAC ("With meta"); (2) Run RAC using the meta-policy without network initialization ("With meta (no init)"); (3) Run RAC with a single regularization based on behavior policy without network initialization, i.e., $\alpha = 1$ ("With beha only"); (4) Run COMBO ("No regul."). As shown in Figure 9(b), "With meta" achieves the best performance and improves significantly over "No regul." and "With beha only", i.e., learning alone without any guidance of meta-policy, which implies that the

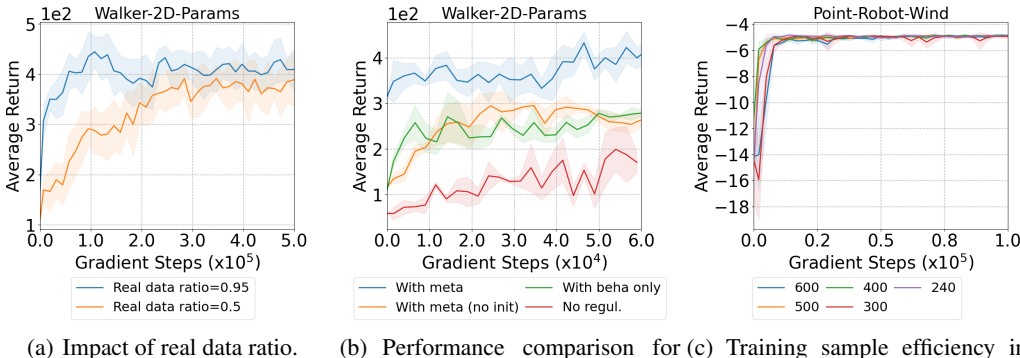

(a) Impact of real data ratio.

(b) Performance comparison for unseen tasks.

(c) Training sample efficiency in Point-Robot-Wind.

Figure 9: Ablation study of MerPO.

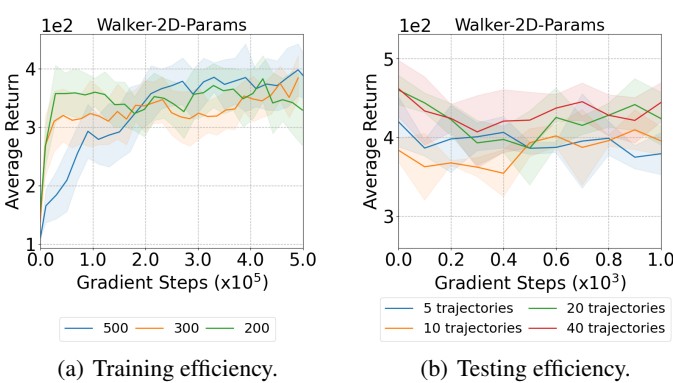

(a) Training efficiency.

(b) Testing efficiency.

Figure 10: Sample efficiency.

learnt meta-policy can efficiently guide the exploration of out-of-distribution state-actions. Without network initialization, "With meta (no init)" and "With beha only" achieve similar performance because good offline dataset is considered here. Such a result is also consistent with Figure 8(d).

We evaluate the testing performance of MerPO, by changing sample size of all tasks. Figure 10(a) shows that the performance of MerPO is stable even if we decrease the number of trajectories for each task to be around 200. In contrast, the number of trajectories collected in other baselines is of the order $10^3$. Figure 10(b) illustrates the testing sample efficiency of MerPO, by evaluating the performance at new offline tasks under different sample sizes. Clearly, a good task-specific policy can be quickly adapted at a new task even with 5 trajectories (1000 samples) of offline data. We also evaluate the training sample efficiency of MerPO in Point-Robot-Wind. As shown in Figure 9(c) the performance of MerPO is stable even if we decrease the number of trajectories for each task to be around 200.

### A.2.4 MORE COMPARISON BETWEEN FOCAL AND COMBO

Following the setup as in Figure 1, we compare the performance between FOCAL and COMBO in two more environments: Half-Cheetah-Fwd-Back and Ant-Fwd-Back. As shown in Figure 11, although FOCAL performs better than COMBO on the task with a bad-quality dataset, it is outperformed by COMBO on the task with a good-quality dataset. This further confirms the observation made in Figure 1.

### A.3 ALGORITHMS

We include the details of MerPO in Algorithm 2.

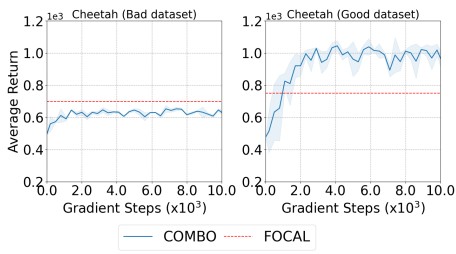
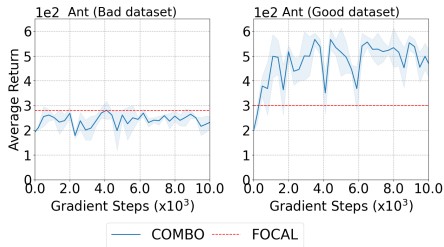

(a) Performance comparison in Half-Cheetah-Fwd-Back.

(b) Performance comparison in Ant-Fwd-Back.

Figure 11: FOCAL vs. COMBO.

---

**Algorithm 2** Regularized policy optimization for model-based offline Meta-RL (MerPO)

---

1: Initialize the dynamics, actor and critic for each task, and initialize the meta-model and the meta-policy;
2: **for** $k = 1, 2, ...$ **do**
3:     **for** each training task $\mathcal{M}_n$ **do**
4:         Solve the following problem with gradient descent for $h$ steps to compute the dynamics model $\widehat{T}_{\theta_n^k}$ based on the offline dataset $\mathcal{D}_i$:
$$\min_{\theta_n} \quad \mathbb{E}_{(s,a,s') \sim \mathcal{D}_n}[\log \widehat{T}_{\theta_n}(s'|s,a)] + \eta \|\theta_n - \phi_m(k)\|_2^2;$$
5:     **end for**
6:     Update $\phi_m(k+1) = \phi_m(k) - \xi_1[\phi_m(k) - \frac{1}{N}\sum_{n=1}^{N}\theta_n^k]$;
7: **end for**
8: Quickly obtain the estimated dynamics model $\widehat{T}_n$ for each training task by solving Eq. (4) with $t$ steps gradient descent;
9: **for** $k = 1, 2, ...$ **do**
10:     **for** each training task $\mathcal{M}_n$ **do**
11:         **for** $j = 1, ..., J$ **do**
12:             Perform model rollouts with $\widehat{T}_n$ starting from states in $\mathcal{D}_n$ and add model rollouts to $\mathcal{D}_{model}^n$;
13:             Policy evaluation by recursively solving Eq. (1) using data from $\mathcal{D}_n \cup \mathcal{D}_{model}^n$;
14:             Given the meta-policy $\pi_c^k$, improve policy $\pi_n^k$ by solving Eq. (7);
15:         **end for**
16:     **end for**
17:     Given the learnt policy $\pi_n^k$ for each task, update the meta-policy $\pi_c^{k+1}$ by solving Eq. (10) with one step gradient descent;
18: **end for**

---

## B  PRELIMINARIES

For ease of exposition, let $T_{\mathcal{M}}$ and $r_{\mathcal{M}}$ denote the dynamics and reward function of the underlying MDP $\mathcal{M}$, $T_{\overline{\mathcal{M}}}$ and $r_{\overline{\mathcal{M}}}$ denote the dynamics and reward function of the empirical MDP $\overline{\mathcal{M}}$ induced by the dataset $\mathcal{D}$, and $T_{\widehat{\mathcal{M}}}$ and $r_{\widehat{\mathcal{M}}}$ denote the dynamics and reward function of the learnt MDP $\widehat{\mathcal{M}}$. To prevent any trivial bound with $\infty$ values, we assume that the cardinality of a state-action pair in the dataset $\mathcal{D}$, i.e., $|\mathcal{D}(s,a)|$, in the denominator, is non-zero, by setting $|\mathcal{D}(s,a)|$ to be a small value less than 1 when $(s,a) \notin \mathcal{D}$.

Following the same line as in (Kumar et al., 2020; Yu et al., 2021b), we make the following standard assumption on the concentration properties of the reward and dynamics for the empirical MDP $\overline{\mathcal{M}}$ to characterize the sampling error.

**Assumption 1.** *For any $(s,a) \in \mathcal{M}$, the following inequalities hold with probability $1 - \delta$:*
$$\|T_{\overline{\mathcal{M}}}(s'|s,a) - T_{\mathcal{M}}(s'|s,a)\|_1 \leq \frac{C_{T,\delta}}{\sqrt{|\mathcal{D}(s,a)|}}; \quad |r_{\overline{\mathcal{M}}}(s,a) - r_{\mathcal{M}}| \leq \frac{C_{r,\delta}}{\sqrt{|\mathcal{D}(s,a)|}}$$
*where $C_{T,\delta}$ and $C_{r,\delta}$ are some constants depending on $\delta$ via a $\sqrt{\log(1/\delta)}$ dependency.*

Based on Assumption 1, we can bound the estimation error induced by the empirical Bellman backup operator for any $(s, a) \in \mathcal{M}$:

$$\left| \mathcal{B}_{\overline{\mathcal{M}}}^\pi \widehat{Q}^k(s, a) - \mathcal{B}_{\mathcal{M}}^\pi \widehat{Q}^k(s, a) \right|$$

$$= \left| r_{\overline{\mathcal{M}}}(s, a) - r_{\mathcal{M}}(s, a) + \gamma \sum_{s'} (T_{\overline{\mathcal{M}}}(s'|s, a) - T_{\mathcal{M}}(s'|s, a)) \mathbb{E}_{\pi(a'|s')}[\widehat{Q}^k(s', a')] \right|$$

$$\leq |r_{\overline{\mathcal{M}}}(s, a) - r_{\mathcal{M}}(s, a)| + \gamma \left| \sum_{s'} (T_{\overline{\mathcal{M}}}(s'|s, a) - T_{\mathcal{M}}(s'|s, a)) \mathbb{E}_{\pi(a'|s')}[\widehat{Q}^k(s', a')] \right|$$

$$\leq \frac{C_{r,\delta}}{\sqrt{|\mathcal{D}(s, a)|}} + \gamma \|T_{\overline{\mathcal{M}}}(s'|s, a) - T_{\mathcal{M}}(s'|s, a)\|_1 \|\mathbb{E}_{\pi(a'|s')}[\widehat{Q}^k(s', a')]\|_\infty$$

$$\leq \frac{C_{r,\delta} + \gamma C_{T,\delta} R_{max}/(1 - \gamma)}{\sqrt{|\mathcal{D}(s, a)|}}$$

$$= \frac{((1 - \gamma) C_{r,\delta}/R_{max} + \gamma C_{T,\delta}) R_{max}}{(1 - \gamma)\sqrt{|\mathcal{D}(s, a)|}}$$

$$\leq \frac{(C_{r,\delta}/R_{max} + C_{T,\delta}) R_{max}}{(1 - \gamma)\sqrt{|\mathcal{D}(s, a)|}} \triangleq \frac{C_{r,T,\delta} R_{max}}{(1 - \gamma)\sqrt{|\mathcal{D}(s, a)|}}.$$

Similarly, we can bound the difference between the Bellman backup induced by the learnt MDP $\widehat{\mathcal{M}}$ and the underlying Bellman backup:

$$\left| \mathcal{B}_{\widehat{\mathcal{M}}}^\pi \widehat{Q}^k(s, a) - \mathcal{B}_{\mathcal{M}}^\pi \widehat{Q}^k(s, a) \right|$$

$$\leq |r_{\widehat{\mathcal{M}}}(s, a) - r_{\mathcal{M}}(s, a)| + \frac{\gamma R_{max}}{1 - \gamma} D_{tv}(T_{\widehat{\mathcal{M}}}, T_{\mathcal{M}})$$

where $D_{tv}(T_{\widehat{\mathcal{M}}}, T_{\mathcal{M}})$ is the total-variation distance between $T_{\widehat{\mathcal{M}}}$ and $T_{\mathcal{M}}$.

For any two MDPs, $\mathcal{M}_1$ and $\mathcal{M}_2$, with the same state space, action space and discount factor $\gamma$, and a given fraction $f \in (0, 1)$, define the $f$-interpolant MDP $\mathcal{M}_f$ as the MDP with dynamics: $T_{\mathcal{M}_f} = f T_{\mathcal{M}_1} + (1 - f) T_{\mathcal{M}_2}$ and reward function: $r_{\mathcal{M}_f} = f r_{\mathcal{M}_1} + (1 - f) r_{\mathcal{M}_2}$, which has the same state space, action space and discount factor with $\mathcal{M}_1$ and $\mathcal{M}_2$. Let $T^\pi$ be the transition matrix on state-action pairs induced by a stationary policy $\pi$, i.e.,

$$T^\pi = T(s'|s, a)\pi(a'|s').$$

To prove the main result, we first restate the following lemma from (Yu et al., 2021b) to be used later.

**Lemma 1.** *For any policy $\pi$, its returns in any MDP $\mathcal{M}$, denoted by $J(\mathcal{M}, \pi)$, and in $\mathcal{M}_f$, denoted by $J(\mathcal{M}_1, \mathcal{M}_2, f, \pi)$, satisfy the following:*

$$J(\mathcal{M}, \pi) - \eta \leq J(\mathcal{M}_1, \mathcal{M}_2, f, \pi) \leq J(\mathcal{M}, \pi) + \eta$$

*where*

$$\eta = \frac{2\gamma(1 - f)}{(1 - \gamma)^2} R_{max} D_{tv}(T_{\mathcal{M}_2}, T_{\mathcal{M}}) + \frac{\gamma f}{1 - \gamma} |\mathbb{E}_{d_{\mathcal{M}}^\pi}[(T_{\mathcal{M}}^\pi - T_{\mathcal{M}_1}^\pi) Q_{\mathcal{M}}^\pi]|$$

$$+ \frac{f}{1 - \gamma} \mathbb{E}_{s, a \sim d_{\mathcal{M}}^\pi}[|r_{\mathcal{M}_1}(s, a) - r_{\mathcal{M}}(s, a)|] + \frac{1 - f}{1 - \gamma} \mathbb{E}_{s, a \sim d_M^\pi}[|r_{\mathcal{M}_2}(s, a) - r_{\mathcal{M}}(s, a)|].$$

Lemma 1 characterizes the relationship between policy returns in different MDPs in terms of the corresponding reward difference and dynamics difference.

## C  PROOF OF THEOREM 1

Let $d(s, a) := d_{\mathcal{M}}^{\pi_\beta}(s, a)$. In the setting without function approximation, by setting the derivation of Equation Eq. (1) to 0, we have that

$$\widehat{Q}^{k+1}(s, a) = \widehat{\mathcal{B}}^\pi \widehat{Q}^k(s, a) - \beta \frac{\rho(s, a) - d(s, a)}{d_f(s, a)}.$$

Denote $\nu(\rho, f) = \mathbb{E}_\rho\left[\frac{\rho(s,a)-d(s,a)}{d_f(s,a)}\right]$ as the expected penalty on the Q-value. It can be shown (Yu et al., 2021b) that $\nu(\rho, f) \geq 0$ and increases with $f$, for any $\rho$ and $f \in (0,1)$. Then, RAC optimizes the return of a policy in a $f$-interpolant MDP induced by the empirical MDP $\overline{\mathcal{M}}$ and the learnt MDP $\widehat{\mathcal{M}}$, which is regularized by both the behavior policy $\pi_\beta$ and the meta-policy $\pi_c$:

$$\max_\pi \quad J(\overline{\mathcal{M}}, \widehat{\mathcal{M}}, f, \pi) - \beta\frac{\nu(\rho^\pi, f)}{1-\gamma} - \lambda\alpha D(\pi, \pi_\beta) - \lambda(1-\alpha)D(\pi, \pi_c). \tag{12}$$

Denote $\pi_o$ as the solution to the above optimization problem. Based on Lemma 1, we can first characterize the return of the learnt policy $\pi_o$ in the underlying MDP $\mathcal{M}$ in terms of its return in the $f$-interpolant MDP:

$$J(M, \pi_o) + \eta_1 \geq J(\overline{\mathcal{M}}, \widehat{\mathcal{M}}, f, \pi_o) \tag{13}$$

where

$$\eta_1 = \frac{2\gamma(1-f)}{(1-\gamma)^2}R_{max}D_{tv}(T_{\widehat{\mathcal{M}}}, T_{\mathcal{M}}) + \frac{\gamma f}{1-\gamma}|\mathbb{E}_{d_{\mathcal{M}}^{\pi_o}}[(T_{\mathcal{M}}^{\pi_o} - T_{\overline{\mathcal{M}}}^{\pi_o})Q_{\mathcal{M}}^{\pi_o}]|$$

$$+ \frac{f}{1-\gamma}\mathbb{E}_{s,a\sim d_{\mathcal{M}}^{\pi_o}}[|r_{\overline{\mathcal{M}}}(s,a) - r_{\mathcal{M}}(s,a)|] + \frac{1-f}{1-\gamma}\mathbb{E}_{s,a\sim d_{\mathcal{M}}^{\pi_o}}[|r_{\widehat{\mathcal{M}}}(s,a) - r_{\mathcal{M}}(s,a)|]$$

$$\leq \frac{2\gamma(1-f)}{(1-\gamma)^2}R_{max}D_{tv}(T_{\widehat{\mathcal{M}}}, T_{\mathcal{M}}) + \frac{\gamma^2 f C_{T,\delta}R_{max}}{(1-\gamma)^2}\mathbb{E}_{s\sim d_{\mathcal{M}}^{\pi_o}(s)}\left[\sqrt{\frac{|A|}{|\mathcal{D}(s)|}}\sqrt{D_{CQL}(\pi_o, \pi_\beta)(s)+1}\right]$$

$$+ \frac{C_{r,\delta}}{1-\gamma}\mathbb{E}_{s,a\sim d_{\mathcal{M}}^{\pi_o}}\left[\frac{1}{\sqrt{|\mathcal{D}(s,a)|}}\right] + \frac{1}{1-\gamma}\mathbb{E}_{s,a\sim d_{\mathcal{M}}^{\pi_o}}[|r_{\widehat{\mathcal{M}}}(s,a) - r_{\mathcal{M}}(s,a)|]$$

$$\triangleq \eta_1^c.$$

Note that the inequality above holds because the following is true for the empirical MDP $\overline{\mathcal{M}}$ (Kumar et al., 2020):

$$|\mathbb{E}_{d_{\overline{\mathcal{M}}}^\pi}[(T_{\mathcal{M}}^\pi - T_{\overline{\mathcal{M}}}^\pi)Q_{\mathcal{M}}^\pi]| \leq \frac{\gamma C_{T,\delta}R_{max}}{1-\gamma}\mathbb{E}_{s\sim d_{\mathcal{M}}^\pi(s)}\left[\sqrt{\frac{|A|}{|\mathcal{D}(s)|}}\sqrt{D_{CQL}(\pi, \pi_\beta)(s)+1}\right]$$

for $D_{CQL}(\pi_1, \pi_2)(s) := \sum_a \pi_1(a|s)\left(\frac{\pi_1(a|s)}{\pi_2(a|s)} - 1\right)$.

## C.1 SAFE IMPROVEMENT OVER $\pi_c$

We first show that the learnt policy offers safe improvement over the meta-policy $\pi_c$. Following the same line as in Eq. (13), we next bound the return of the meta-policy $\pi_c$ in the underlying MDP $\mathcal{M}$ from above, in terms of its return in the $f$-interpolant MDP:

$$J(\overline{\mathcal{M}}, \widehat{\mathcal{M}}, f, \pi_c) \geq J(M, \pi_c) - \eta_2$$

where

$$\eta_2 \leq \frac{2\gamma(1-f)}{(1-\gamma)^2}R_{max}D_{tv}(T_{\widehat{\mathcal{M}}}, T_{\mathcal{M}}) + \frac{\gamma^2 f C_{T,\delta}R_{max}}{(1-\gamma)^2}\mathbb{E}_{s\sim d_{\mathcal{M}}^{\pi_c}(s)}\left[\sqrt{\frac{|A|}{|\mathcal{D}(s)|}}\sqrt{D_{CQL}(\pi_c, \pi_\beta)(s)+1}\right]$$

$$+ \frac{C_{r,\delta}}{1-\gamma}\mathbb{E}_{s,a\sim d_{\mathcal{M}}^{\pi_c}}\left[\frac{1}{\sqrt{|\mathcal{D}(s,a)|}}\right] + \frac{1}{1-\gamma}\mathbb{E}_{s,a\sim d_{\mathcal{M}}^{\pi_c}}[|r_{\widehat{\mathcal{M}}}(s,a) - r_{\mathcal{M}}(s,a)|]$$

$$\triangleq \eta_2^c.$$

It follows that

$$J(\mathcal{M}, \pi_o) + \eta_1^c - \beta\frac{\nu(\rho^{\pi_o}, f)}{1-\gamma} - \lambda\alpha D(\pi_o, \pi_\beta) - \lambda(1-\alpha)D(\pi_o, \pi_c)$$

$$\geq J(\overline{\mathcal{M}}, \widehat{\mathcal{M}}, f, \pi_o) - \beta\frac{\nu(\rho^{\pi_o}, f)}{1-\gamma} - \lambda\alpha D(\pi_o, \pi_\beta) - \lambda(1-\alpha)D(\pi_o, \pi_c)$$

$$\geq J(\overline{\mathcal{M}}, \widehat{\mathcal{M}}, f, \pi_c) - \beta\frac{\nu(\rho^{\pi_c}, f)}{1-\gamma} - \lambda\alpha D(\pi_c, \pi_\beta)$$

$$\geq J(\mathcal{M}, \pi_c) - \eta_2^c - \beta\frac{\nu(\rho^{\pi_c}, f)}{1-\gamma} - \lambda\alpha D(\pi_c, \pi_\beta),$$

where the second inequality is true because $\pi_o$ is the solution to Eq. (12). This gives us a lower bound on $J(\mathcal{M}, \pi_o)$ in terms of $J(\mathcal{M}, \pi_c)$:

$$J(\mathcal{M}, \pi_o) \geq J(\mathcal{M}, \pi_c) - \eta_1^c - \eta_2^c + \frac{\beta}{1-\gamma}[\nu(\rho^{\pi_o}, f) - \nu(\rho^{\pi_c}, f)]$$
$$+ \lambda\alpha D(\pi_o, \pi_\beta) + \lambda(1-\alpha)D(\pi_o, \pi_c) - \lambda\alpha D(\pi_c, \pi_\beta).$$

It is clear that $\eta_1^c$ and $\eta_2^c$ are independent to $\beta$ and $\lambda$. To show the performance improvement of $\pi_o$ over the meta-policy $\pi_c$, it suffices to guarantee that for appropriate choices of $\beta$ and $\lambda$,

$$\Delta_c = \lambda\alpha D(\pi_o, \pi_\beta) + \lambda(1-\alpha)D(\pi_o, \pi_c) - \lambda\alpha D(\pi_c, \pi_\beta) + \frac{\beta}{1-\gamma}[\nu(\rho^{\pi_o}, f) - \nu(\rho^{\pi_c}, f)] > 0.$$

To this end, the following lemma first provides an upper bound on $|\nu(\rho^{\pi_o}, f) - \nu(\rho^{\pi_c}, f)|$:

**Lemma 2.** *There exist some positive constants $L_1$ and $L_2$ such that*
$$|\nu(\rho^{\pi_o}, f) - \nu(\rho^{\pi_c}, f)| \leq 2(L_1 + L_2)D_{tv}(\rho^{\pi_o}(s,a)||\rho^{\pi_c}(s,a)).$$

*Proof.* First, we have that
$$|\nu(\rho^{\pi_o}, f) - \nu(\rho^{\pi_c}, f)|$$

$$= \left| \mathbb{E}_{\rho^{\pi_o}}\left[\frac{\rho^{\pi_o}(s,a) - d(s,a)}{fd(s,a) + (1-f)\rho^{\pi_o}(s,a)}\right] - \mathbb{E}_{\rho^{\pi_c}}\left[\frac{\rho^{\pi_c}(s,a) - d(s,a)}{fd(s,a) + (1-f)\rho^{\pi_c}(s,a)}\right] \right|$$

$$= \left| \sum_{(s,a)}\left[\rho^{\pi_o}(s,a)\frac{\rho^{\pi_o}(s,a) - d(s,a)}{fd(s,a) + (1-f)\rho^{\pi_o}(s,a)} - \rho^{\pi_c}(s,a)\frac{\rho^{\pi_c}(s,a) - d(s,a)}{fd(s,a) + (1-f)\rho^{\pi_c}(s,a)}\right] \right|$$

$$\leq \left| \sum_{(s,a)}\left[\rho^{\pi_o}(s,a)\frac{\rho^{\pi_o}(s,a) - d(s,a)}{fd(s,a) + (1-f)\rho^{\pi_o}(s,a)} - \rho^{\pi_c}(s,a)\frac{\rho^{\pi_o}(s,a) - d(s,a)}{fd(s,a) + (1-f)\rho^{\pi_o}(s,a)}\right] \right|$$

$$+ \left| \sum_{(s,a)}\left[\rho^{\pi_c}(s,a)\frac{\rho^{\pi_o}(s,a) - d(s,a)}{fd(s,a) + (1-f)\rho^{\pi_o}(s,a)} - \rho^{\pi_c}(s,a)\frac{\rho^{\pi_c}(s,a) - d(s,a)}{fd(s,a) + (1-f)\rho^{\pi_c}(s,a)}\right] \right|$$

$$= \left| \sum_{(s,a)}[\rho^{\pi_o}(s,a) - \rho^{\pi_c}(s,a)]\frac{\rho^{\pi_o}(s,a) - d(s,a)}{fd(s,a) + (1-f)\rho^{\pi_o}(s,a)} \right|$$

$$+ \left| \sum_{(s,a)}\rho^{\pi_c}(s,a)\left[\frac{\rho^{\pi_o}(s,a) - d(s,a)}{fd(s,a) + (1-f)\rho^{\pi_o}(s,a)} - \frac{\rho^{\pi_c}(s,a) - d(s,a)}{fd(s,a) + (1-f)\rho^{\pi_c}(s,a)}\right] \right|$$

$$\leq \sum_{(s,a)}|\rho^{\pi_o}(s,a) - \rho^{\pi_c}(s,a)|\left|\frac{\rho^{\pi_o}(s,a) - d(s,a)}{fd(s,a) + (1-f)\rho^{\pi_o}(s,a)}\right|$$

$$+ \sum_{(s,a)}\rho^{\pi_c}(s,a)\left|\frac{\rho^{\pi_o}(s,a) - d(s,a)}{fd(s,a) + (1-f)\rho^{\pi_o}(s,a)} - \frac{\rho^{\pi_c}(s,a) - d(s,a)}{fd(s,a) + (1-f)\rho^{\pi_c}(s,a)}\right|.$$

First, observe that for the term $\left|\frac{\rho^{\pi_o}(s,a) - d(s,a)}{fd(s,a) + (1-f)\rho^{\pi_o}(s,a)}\right|$,

- If $\rho^{\pi_o}(s,a) \geq d(s,a)$, then
$$\left|\frac{\rho^{\pi_o}(s,a) - d(s,a)}{fd(s,a) + (1-f)\rho^{\pi_o}(s,a)}\right|$$
$$\leq \left|\frac{\rho^{\pi_o}(s,a)}{fd(s,a) + (1-f)\rho^{\pi_o}(s,a)}\right| \leq \left|\frac{\rho^{\pi_o}(s,a)}{(1-f)\rho^{\pi_o}(s,a)}\right| = \frac{1}{1-f}.$$

- If $\rho^{\pi_o}(s,a) < d(s,a)$, then
$$\left|\frac{\rho^{\pi_o}(s,a) - d(s,a)}{fd(s,a) + (1-f)\rho^{\pi_o}(s,a)}\right| \leq \left|\frac{d(s,a)}{fd(s,a) + (1-f)\rho^{\pi_o}(s,a)}\right| \leq \left|\frac{d(s,a)}{fd(s,a)}\right| = \frac{1}{f}.$$

Therefore,

$$\left|\frac{\rho^{\pi_o}(s,a) - d(s,a)}{fd(s,a) + (1-f)\rho^{\pi_o}(s,a)}\right| \le \max\left\{\frac{1}{f}, \frac{1}{1-f}\right\} \triangleq L_1.$$

Next, for the term $\left|\frac{\rho^{\pi_o}(s,a)-d(s,a)}{fd(s,a)+(1-f)\rho^{\pi_o}(s,a)} - \frac{\rho^{\pi_c}(s,a)-d(s,a)}{fd(s,a)+(1-f)\rho^{\pi_c}(s,a)}\right|$, consider the function $g(x) = \frac{x-d}{fd+(1-f)x}$ for $x \in [0,1]$. Clearly, when $d(s,a) = 0$,

$$\left|\frac{\rho^{\pi_o}(s,a) - d(s,a)}{fd(s,a) + (1-f)\rho^{\pi_o}(s,a)} - \frac{\rho^{\pi_c}(s,a) - d(s,a)}{fd(s,a) + (1-f)\rho^{\pi_c}(s,a)}\right| = 0.$$

For any $(s,a)$ that $d(s,a) > 0$, it can be shown that $g(x)$ is continuous and has bounded gradient, i.e., $|\nabla g(x)| \le \frac{1}{f^2 d} \triangleq L_2$. Hence, it follows that

$$\left|\frac{\rho^{\pi_o}(s,a) - d(s,a)}{fd(s,a) + (1-f)\rho^{\pi_o}(s,a)} - \frac{\rho^{\pi_c}(s,a) - d(s,a)}{fd(s,a) + (1-f)\rho^{\pi_c}(s,a)}\right| \le L_2|\rho^{\pi_o}(s,a) - \rho^{\pi_c}(s,a)|.$$

Therefore, we can conclude that

$$|\nu(\rho^{\pi_o}, f) - \nu(\rho^{\pi_c}, f)|$$

$$\le L_1 \sum_{(s,a)} |\rho^{\pi_o}(s,a) - \rho^{\pi_c}(s,a)| + L_2 \sum_{(s,a)} \rho^{\pi_c}(s,a)|\rho^{\pi_o}(s,a) - \rho^{\pi_c}(s,a)|$$

$$\le (L_1 + L_2) \sum_{s,a} |\rho^{\pi_o}(s,a) - \rho^{\pi_c}(s,a)|$$

$$= 2(L_1 + L_2) D_{tv}(\rho^{\pi_o}(s,a)||\rho^{\pi_c}(s,a)).$$

$\square$

Recall that

$$\rho^{\pi_o}(s,a) = d_{\widehat{\mathcal{M}}}^{\pi_o}(s)\pi_o(a|s), \quad \rho^{\pi_c}(s,a) = d_{\widehat{\mathcal{M}}}^{\pi_c}(s)\pi_c(a|s),$$

which denote the marginal state-action distributions by rolling out $\pi_o$ and $\pi_c$ in the learnt model $\widehat{\mathcal{M}}$, respectively. Lemma 2 gives an upper bound on the difference between the expected penalties induced under $\pi_o$ and $\pi_c$, with regard to the difference between the marginal state-action distributions. Next, we need to characterize the relationship between the marginal state-action distribution difference and the corresponding policy distance, which is captured in the following lemma.

**Lemma 3.** *Let $D(\pi_1||\pi_2) = \max_s D_{tv}(\pi_1||\pi_2)$ denote the maximum total-variation distance between two policies $\pi_1$ and $\pi_2$. Then, we can have that*

$$D_{tv}(\rho^{\pi_o}(s,a)||\rho^{\pi_c}(s,a)) \le \frac{1}{1-\gamma} \max_s D_{tv}(\pi_o(a|s)||\pi_c(a|s)).$$

*Proof.* Note that

$$D_{tv}(\rho^{\pi_o}(s,a)||\rho^{\pi_c}(s,a)) \le (1-\gamma) \sum_{t=0}^{\infty} \gamma^t D_{tv}(\rho_t^{\pi_o}(s,a)||\rho_t^{\pi_c}(s,a)).$$

It then suffices to bound the state-action marginal difference at time $t$. Since both state-action marginals here correspond to rolling out $\pi_o$ and $\pi_c$ in the same MDP $\widehat{\mathcal{M}}$, based on Lemma B.1 and B.2 in (Janner et al., 2019), we can obtain that

$$D_{tv}(\rho_t^{\pi_o}(s,a)||\rho_t^{\pi_c}(s,a))$$

$$\le D_{tv}(\rho_t^{\pi_o}(s)||\rho_t^{\pi_c}(s)) + \max_s D_{tv}(\pi_o(a|s)||\pi_c(a|s))$$

$$\le t \max_s D_{tv}(\pi_o(a|s)||\pi_c(a|s)) + \max_s D_{tv}(\pi_o(a|s)||\pi_c(a|s))$$

$$= (t+1) \max_s D_{tv}(\pi_o(a|s)||\pi_c(a|s)),$$

which indicates that

$$D_{tv}(\rho^{\pi_o}(s,a)||\rho^{\pi_c}(s,a)) \le (1-\gamma) \sum_{t=0}^{\infty} \gamma^t (t+1) \max_s D_{tv}(\pi_o(a|s)||\pi_c(a|s))$$

$$= \frac{1}{1-\gamma} \max_s D_{tv}(\pi_o(a|s)||\pi_c(a|s)).$$

$\square$

Building on Lemma 2 and Lemma 3, we can show that

$$|\nu(\rho^{\pi_o}, f) - \nu(\rho^{\pi_c}, f)| \leq \frac{2(L_1 + L_2)}{1 - \gamma} \max_s D_{tv}(\pi_o(a|s)||\pi_c(a|s))$$

$$\triangleq C \max_s D_{tv}(\pi_o(a|s)||\pi_c(a|s)).$$

Let $D(\cdot, \cdot) = \max_s D_{tv}(\cdot||\cdot)$. It is clear that for $\lambda \geq \lambda_0$ where $\lambda_0 > \frac{C\beta}{(1-\gamma)(1-2\alpha)}$ and $\alpha < \frac{1}{2}$,

$$\Delta_c = \lambda\alpha D(\pi_o, \pi_\beta) + \lambda(1 - \alpha)D(\pi_o, \pi_c) - \lambda\alpha D(\pi_c, \pi_\beta) + \frac{\beta}{1 - \gamma}[\nu(\rho^{\pi_o}, f) - \nu(\rho^{\pi_c}, f)]$$

$$= \lambda\alpha D(\pi_o, \pi_\beta) + \lambda\alpha D(\pi_o, \pi_c) - \lambda\alpha D(\pi_c, \pi_\beta) + \lambda(1 - 2\alpha)D(\pi_o, \pi_c)$$

$$+ \frac{\beta}{1 - \gamma}[\nu(\rho^{\pi_o}, f) - \nu(\rho^{\pi_c}, f)]$$

$$\geq \lambda(1 - 2\alpha)D(\pi_o, \pi_c) + \frac{\beta}{1 - \gamma}[\nu(\rho^{\pi_o}, f) - \nu(\rho^{\pi_c}, f)]$$

$$\geq \lambda(1 - 2\alpha)D(\pi_o, \pi_c) - \frac{C\beta}{1 - \gamma}D(\pi_o, \pi_c)$$

$$= (\lambda - \lambda_0)(1 - 2\alpha)D(\pi_o, \pi_c) + \left[\lambda_0(1 - 2\alpha) - \frac{C\beta}{1 - \gamma}\right]D(\pi_o, \pi_c) > 0.$$

In a nutshell, we can conclude that with probability $1 - \delta$

$$J(\mathcal{M}, \pi_o) \geq J(\mathcal{M}, \pi_c) \underbrace{-\eta_1^c - \eta_2^c}_{(a)} + \underbrace{(\lambda - \lambda_0)(1 - 2\alpha)D(\pi_o, \pi_c)}_{(b)} + \underbrace{\left[\lambda_0(1 - 2\alpha) - \frac{C\beta}{1 - \gamma}\right]D(\pi_o, \pi_c)}_{(c)},$$

where (a) depends on $\delta$ but is independent to $\lambda$, (b) is positive and increases with $\lambda$, and (c) is positive. This implies that an appropriate choice of $\lambda$ will make term (b) large enough to counteract term (a) and lead to the performance improvement over the meta-policy $\pi_c$:

$$J(\mathcal{M}, \pi_o) \geq J(\mathcal{M}, \pi_c) + \xi_1$$

where $\xi_1 \geq 0$.

## C.2 SAFE IMPROVEMENT OVER $\pi_\beta$

Next, we show that the learnt policy $\pi_o$ achieves safe improvement over the behavior policy $\pi_\beta$. Based on Lemma 1, we have

$$J(\mathcal{M}_1, \mathcal{M}_2, f, \pi_\beta) \geq J(\mathcal{M}, \pi_\beta) - \eta_3$$

where

$$\eta_3 = \frac{2\gamma(1 - f)}{(1 - \gamma)^2}R_{max}D_{tv}(T_{\widehat{\mathcal{M}}}, T_{\mathcal{M}}) + \frac{\gamma f}{1 - \gamma}|\mathbb{E}_{d_{\mathcal{M}}^{\pi_\beta}}[(T_{\mathcal{M}}^{\pi_\beta} - T_{\overline{\mathcal{M}}}^{\pi_\beta})Q_{\mathcal{M}}^{\pi_\beta}]|$$

$$+ \frac{f}{1 - \gamma}\mathbb{E}_{s,a \sim d_{\mathcal{M}}^{\pi_\beta}}[|r_{\overline{\mathcal{M}}}(s, a) - r_{\mathcal{M}}(s, a)|] + \frac{1 - f}{1 - \gamma}\mathbb{E}_{s,a \sim d_{\mathcal{M}}^{\pi_\beta}}[|r_{\widehat{\mathcal{M}}}(s, a) - r_{\mathcal{M}}(s, a)|]$$

$$\leq \frac{2\gamma(1 - f)}{(1 - \gamma)^2}R_{max}D_{tv}(T_{\widehat{\mathcal{M}}}, T_{\mathcal{M}}) + \frac{\gamma^2 f C_{T,\delta}R_{max}}{(1 - \gamma)^2}\mathbb{E}_{s \sim d_{\mathcal{M}}^{\pi_\beta}(s)}\left[\sqrt{\frac{|A|}{|\mathcal{D}(s)|}}\right]$$

$$+ \frac{C_{r,\delta}}{1 - \gamma}\mathbb{E}_{s,a \sim d_{\mathcal{M}}^{\pi_\beta}}\left[\frac{1}{\sqrt{|\mathcal{D}(s,a)|}}\right] + \frac{1}{1 - \gamma}\mathbb{E}_{s,a \sim d_{\mathcal{M}}^{\pi_\beta}}[|r_{\widehat{\mathcal{M}}}(s, a) - r_{\mathcal{M}}(s, a)|]$$

$$\triangleq \eta_3^\beta.$$

Therefore, it follows that

$$J(\mathcal{M}, \pi_o) + \eta_1^c - \beta \frac{\nu(\rho^{\pi_o}, f)}{1 - \gamma} - \lambda \alpha D(\pi_o, \pi_\beta) - \lambda(1 - \alpha) D(\pi_o, \pi_c)$$

$$\geq J(\overline{\mathcal{M}}, \widehat{\mathcal{M}}, f, \pi_o) - \beta \frac{\nu(\rho^{\pi_o}, f)}{1 - \gamma} - \lambda \alpha D(\pi_o, \pi_\beta) - \lambda(1 - \alpha) D(\pi_o, \pi_c)$$

$$\geq J(\overline{\mathcal{M}}, \widehat{\mathcal{M}}, f, \pi_\beta) - \beta \frac{\nu(\rho^{\pi_\beta}, f)}{1 - \gamma} - \lambda(1 - \alpha) D(\pi_\beta, \pi_c)$$

$$\geq J(\mathcal{M}, \pi_\beta) - \eta_3^\beta - \beta \frac{\nu(\rho^{\pi_\beta}, f)}{1 - \gamma} - \lambda(1 - \alpha) D(\pi_\beta, \pi_c),$$

which indicates that with probability $1 - \delta$

$$J(\mathcal{M}, \pi_c) \geq J(\mathcal{M}, \pi_\beta) - \eta_1^c - \eta_3^\beta + \lambda \alpha D(\pi_o, \pi_\beta) + \lambda(1 - \alpha) D(\pi_o, \pi_c) - \lambda(1 - \alpha) D(\pi_\beta, \pi_c)$$

$$+ \frac{\beta}{1 - \gamma} [\nu(\rho^{\pi_o}, f) - \nu(\rho^{\pi_\beta}, f)],$$

where $\eta_3^\beta$ is some constant that depends on $\delta$ but is independent to $\beta$ and $\lambda$.

To conclude, we can have that with probability $1 - 2\delta$

$$J(\mathcal{M}, \pi_o) \geq \max\{J(\mathcal{M}, \pi_c) + \xi_1, J(\mathcal{M}, \pi_\beta) + \xi_2\}$$

where

$$\xi_1 = -\eta_1^c - \eta_2^c + (\lambda - \lambda_0)(1 - 2\alpha) D(\pi_o, \pi_c) + \left[ \lambda_0(1 - 2\alpha) - \frac{C\beta}{1 - \gamma} \right] D(\pi_o, \pi_c) \qquad (14)$$

and

$$\xi_2 = -\eta_1^c - \eta_3^\beta + \lambda \alpha D(\pi_o, \pi_\beta) + \lambda(1 - \alpha) D(\pi_o, \pi_c) - \lambda(1 - \alpha) D(\pi_\beta, \pi_c) \qquad (15)$$

$$+ \frac{\beta}{1 - \gamma} [\nu(\rho^{\pi_o}, f) - \nu(\rho^{\pi_\beta}, f)]. \qquad (16)$$

Moreover, as we noted earlier, $\xi_1 > 0$ for a suitably selected $\lambda$ and $\alpha < \frac{1}{2}$. For the term $\nu(\rho^{\pi_o}, f) - \nu(\rho^{\pi_\beta}, f)$ in $\xi_2$ where $\nu(\rho^\pi, f)$ is defined as $\mathbb{E}_{\rho^\pi} \left[ \frac{\rho^\pi(s,a) - d(s,a)}{d_f(s,a)} \right]$, as noted in (Yu et al., 2021b), $\nu(\rho^{\pi_\beta}, f)$ is expected to be smaller than $\nu(\rho^{\pi_o}, f)$ in practical scenarios, due to the fact that the dynamics $T_{\widehat{\mathcal{M}}}$ learnt via supervised learning is close to the underlying dynamics $T_{\mathcal{M}}$ on the states visited by the behavior policy $\pi_\beta$. This directly indicates that $d_{\widehat{\mathcal{M}}}^{\pi_\beta}(s, a)$ is close to $d_{\mathcal{M}}^{\pi_\beta}(s, a)$ and $\rho^{\pi_\beta}$ is close to $d(s, a)$. In this case, let

$$\epsilon = \frac{\beta[\nu(\rho^{\pi_o}, f) - \nu(\rho^{\pi_\beta}, f)]}{2\lambda(1 - \gamma) D(\pi_o, \pi_\beta)}.$$

We can show that for $\alpha > \frac{1}{2} - \epsilon$,

$$\Delta_\beta = \lambda \alpha D(\pi_o, \pi_\beta) + \lambda(1 - \alpha) D(\pi_o, \pi_c) - \lambda(1 - \alpha) D(\pi_c, \pi_\beta) + \frac{\beta}{1 - \gamma} [\nu(\rho^{\pi_o}, f) - \nu(\rho^{\pi_\beta}, f)]$$

$$= \lambda \alpha D(\pi_o, \pi_\beta) + \lambda(1 - \alpha) D(\pi_o, \pi_c) - \lambda(1 - \alpha) D(\pi_c, \pi_\beta) + 2\epsilon \lambda D(\pi_o, \pi_\beta)$$

$$= \lambda \left[ (2\epsilon + \alpha) D(\pi_o, \pi_\beta) + (1 - \alpha) D(\pi_o, \pi_c) - (1 - \alpha) D(\pi_c, \pi_\beta) \right]$$

$$> \lambda(1 - \alpha) [D(\pi_o, \pi_\beta) + D(\pi_o, \pi_c) - D(\pi_c, \pi_\beta)]$$

$$> 0,$$

and $\Delta_\beta$ increases with $\lambda$, which implies that

$$J(\mathcal{M}, \pi_o) \geq J(\mathcal{M}, \pi_\beta) + \xi_2 = J(\mathcal{M}, \pi_\beta) - \eta_1^c - \eta_3^\beta + \Delta_\beta > J(\mathcal{M}, \pi_\beta)$$

for an appropriate choice of $\lambda$.

