# OpenReview forum: "Model-Based Offline Meta-Reinforcement Learning with Regularization"
_ICLR.cc/2022/Conference — ICLR 2022 Poster_

### Official Review · Reviewer_zWVS · 2021-10-26

**Correctness:** 4
**Technical Novelty And Significance:** 3
**Empirical Novelty And Significance:** 3
**Recommendation:** 8
**Confidence:** 4

**Main Review:**

This paper proposes a new model-based offline meta-rl algorithm under the actor-critic approach. It is a nice addition to existing algorithms for the study of meta-RL problem. The difficulty of balancing the exploration with meta-policy and exploitation with offline dataset is an interesting observation, and the proposed solution is a reasonable remedy.

The main idea and the corresponding algorithm are well presented. The paper starts with a clear motivation in the introduction. After explaining the background, it points out the limitation of applying online meta-RL approach to the offline setting in section 4.2.1, and proposes its solution to introduce another regularisation around behaviour policy. The full algorithm include many components, meta-learning for dynamics, model based policy optimisation, conservative Q-learning, regularisation with behaviour and meta-policy. I am glad that the authors explain each component clearly and how they interact with each other.

The solution to add a second regularisation to task-specific policy seems simple and intuitive, but it is nice to show the proposed method does improve the performance both theoretically and empirically. The main theory results look reasonable but I haven't checked the detailed derivation in the appendix.
The experiments results are convincing and the ablation study is very helpful to show the contribution of each component.

I have a few questions about the algorithm design:

- CQL already encourages the Q function, and consequently the learned policy, to stay close to the behaviour policy. Why do you think it's not enough and we need another explicit regularisation to pull the policy close to the behaviour policy?

- Instead of regularising task-policy to behaviour policy in Eq 7, we can also consider regularising the meta-policy to behaviour policy in order to prevent the quality of the meta-policy to decrease. Could the authors comment on this option?

- Selecting a proper alpha value is the key to the proposed algorithm. Unfortunately the range provided by Theorem 1 is not computable and it seems hard to find an appropriate value according to the available offline dataset of each new task in practice. While the authors show 0.4 is a good value for all the experiments, I doubt if we can trust it in general. Hoping a constant value of alpha to work for all is counterintuitive because the whole paper is arguing that we should balance the two regularisation according to the quality of the dataset and the meta-policy. So we should adjust the interpolation coefficient accordingly.









**Summary Of The Paper:**

This paper considers the challenge in offline meta-RL problems, particularly, the difficulty with balancing a learned policy between exploring OOD state-actions by following the meta-policy and exploiting the offline dataset. To solve the problem, it starts with proximal meta-RL approach in the online setting, points out the problem of a degraded meta-policy in the offline setting due to lack of feedback from online interactions and proposes to regularize the task policy with additional penalty from deviating from the behaviour policy. It also integrates the model-based offline RL method for single tasks (COMBO) and offline meta-learning for task specific dynamics into its final form (MerPO). This paper provides theoretical analysis on its advantage over behaviour policy and the meta policy. Extensive experiments with ablation show the empirical performance meets its expected behaviour.

**Summary Of The Review:**

The paper has a good motivation and provides a reasonable solution to the problem in offline meta-RL. Theoretical and empirical results support the advantage of the proposed algorithm over recent baseline algorithms.

---

> ### Author Response · Authors · 2021-11-22
> **Reply to Reviewer zWVS (2/2)**
>
> Q3: We should adjust the interpolation coefficient accordingly.
>
> A3: Many thanks for the insightful comments. Following the reviewer's comment, we designed an algorithm with $\alpha$ adaptive to tasks and during the learning process and experimentally showed that such an algorithm does outperform our previous algorithm with fixed $\alpha$. We summarize our result as follows.
>
> As suggested by the reviewer, since the meta-policy $\pi_c$ changes during the learning process and the qualities of the behavior policies vary across different tasks, we adapt $\alpha$ across different iterations and tasks so as to achieve a 'local' balance between the impacts of the meta-policy and the behavior policy. We name the new method as MerPO-Adp. Our design idea obeys the widely-adopted rule of learning conservatively in offline RL (see e.g., Fujimoto et al. 2019, Kumar et al. 2019, Kumar et al. 2020, Yu et al. 2020, Kidambi et al. 2020). Specifically, at each iteration $k$, given the task-policy $\pi_n^k$ for task $n$ and the meta-policy $\pi_c^k$ at iteration $k$, we update $\alpha^k_n$ using one-step gradient descent to minimize the following problem.
>
> $$\min_{\alpha^k_n} (1-\alpha^k_n)[D(\pi^k_{n},\pi_{\beta,n})-D(\pi^k_n,\pi^k_c)], s.t. \alpha^k_n\in[0.1,0.5].$$
>
> The idea is to adapt $\alpha_n^k$ in order to balance between $D(\pi^k_n,\pi_{\beta,n})$ and $D(\pi^k_n,\pi^k_c)$, because our Theorem 1 implies that the safe policy improvement can be achieved when the impacts of the meta-policy and the behavior policy are well balanced. Specifically, at iteration $k$ for each task $n$,  $\alpha^k_n$ is increased when the task-policy $\pi_n^k$ is closer to the meta-policy $\pi^k_c$, and is decreased  when the task-policy is closer to the behavior policy. It is worth noting that $\alpha^k_n$ is constrained in the range $[0.1,0.5]$ as suggested by Theorem 1.
>
> We conduct experiments to evaluate the performance of the new algorithm MerPO-Adp in Section 5.2. Figure 6 indicates that MerPO-Adp can achieve better or comparable performance compared to MerPO with a fixed $\alpha$.

---

> > ### Comment · Reviewer_zWVS · 2021-11-29
> > **Concerns resolved**
> >
> > Thanks for the authors' response and additional experiments. I'm happy to see the new method with adaptive \alpha give a better or comparable performance. My concerns are all resolved.

---

> ### Author Response · Authors · 2021-11-22
> **Reply to Reviewer zWVS (1/2)**
>
> Thank you for your thorough reviews and constructive comments. In our revision of the paper, we added new experiments on adaptive $\alpha$ in section 5.2 and on the comparison between FOCAL and COMBO in the appendix A.2.4, and made various revisions throughout the paper based on all reviewers’ comments. All our changes are highlighted with blue-colored texts. New comments on these changes are very welcome!
>
> Q1: CQL already encourages the Q function, and consequently the learnt policy, to stay close to the behavior policy. Why do you think it's not enough and we need another explicit regularization to pull the policy close to the behavior policy?
>
> A1: Great question! There are two main reasons behind this.
>
> - CQL (or COMBO) implicitly regularizes the learnt task policy to be close to the behavior policy through a conservative policy evaluation. However, without the behavior regularization in the policy improvement, the within-task policy improvement step Eq. (6) regularizes the learnt task policy to be close to the meta-policy in a more explicit and direct way. It is possible that the regularizer with the meta-policy dominates the performance and pulls the task-policy close to the meta-policy. This would lead to a poor task-policy when the quality of the meta-policy is poor, especially in the early stage of meta-training (This can also be seen from Figure 4 about the performance of COMBO-3 with a poor meta-policy). The poor task-policy in turn regularizes the meta-policy update and results in a worse meta-policy. The iterative meta-training process would eventually lead to unsatisfactory performance as illustrated in Figure 3.
>
> - To balance the tradeoff between staying close to the behavior policy and following the meta-policy, it is desirable to explicitly quantify the impact of behavior regularization and model the interaction between the behavior policy and the meta-policy, which however is challenging and not clear through the implicit regularization in CQL. Towards this end, we explicitly introduce a weighted interpolation between two different regularizers based on the behavior policy and the meta-policy for within-task policy improvement. For an appropriate choice of the weight, the intrinsic tradeoff between staying close to the behavior policy and following the meta-policy can be achieved through the weighted interpolation, with the safe improvement guarantee in Theorem 1. We will further investigate this direction in the future work.
>
>
> Q2: Instead of regularizing task-policy to behavior policy in Eq 7, we can also consider regularizing the meta-policy to behavior policy in order to prevent the quality of the meta-policy to decrease. Could the authors comment on this option?
>
> A2: We sincerely thank you for pointing out this interesting direction, and we have the following two thoughts on this direction.
>
> - One of the motivations in offline meta-RL is to learn a meta-policy that could guide the "exploration" of out-of-distribution state-actions beyond the offline datasets (behavior policy), by leveraging the knowledge across similar offline tasks. However, regularizing the meta-policy to stay close to the behavior policy could potentially lead to a meta-policy that is similar to the behavior policy, which consequently hinders the exploration of out-of-distribution state-actions. The fact, that the behavior policies of different tasks have varying qualities and some may be poor, would also pose some challenges to this design.
>
> - Intuitively, regularizing the meta-policy to be close to the behavior policy could alleviate the quality degradation of the meta-policy to some extent, but this deserves more thorough investigation and analysis. In particular, the meta-policy update is now directly regularized by two different regularizers, one based on the learnt task-policy and the other on the behavior policy. It is not clear that how these two regularizers interact with each other and what impact they would eventually have on the performance of the meta-policy. We will continue to investigate this interesting direction in the future.

---

### Official Review · Reviewer_xD1s · 2021-10-28

**Correctness:** 2
**Technical Novelty And Significance:** 2
**Empirical Novelty And Significance:** 2
**Recommendation:** 6
**Confidence:** 3

**Main Review:**

This paper studies an interesting question of offline meta RL and presents promising experimental results. The ablations do a good job of demonstrating that each component of the method contributes to the success of the overall method.

One concern with the paper is that the motivation for the method is rather unclear. This lack of clarity is due in part because the paper does not make specific, falsifiable claims. For example, one question posed is, “Why does the proximal Meta-RL method in Eq. (5) perform poorly in offline Meta-RL, even with conservative policy evaluation?” The paper continues to say that, “A poor meta-policy may have negative impact on the performance” and that, “following the meta-policy may lead to worse performance.” It is always possible that something “may” lead to worse performance, and this does not directly motivate the conclusion that, “it is necessary to balance the tradeoff between exploring with the meta-policy and exploiting the offline dataset, in order to guarantee the performance improvement of new offline tasks.” It would be great for the authors to provide direct evidence that this trade-off is necessary.

Similarly, the results in Figure 1 do not justify the claim that, “Clearly, existing offline Meta-RL fails to generalize equally well over datasets with varied quality.” Figure 1 only looks at one specific offline meta-RL algorithm and on one environment. Making such a general claim would require much more extensive evidence, and I do not see how, even if the claim were true, that would directly motivate the need to develop a method that will “strike the right balance between exploring with the meta-policy and exploiting the offline dataset.”

In short, the paper does not provide evidence that striking *this* balance is the main issue that must be addressed.

Another concern with the paper is that it is a relatively complex combination of existing components (COMBO, proximal meta RL, behavior cloning regularization), and so the onus is on the paper to demonstrate exceedingly good results to justify the complexity of the method. Although the results are positive, the method only mildly improves over FOCAL, and the paper would be strengthened by comparing to MACAW and BOReL, as those methods have been shown to perform well in these environments.

Another concern is that it is unclear how alpha is tuned or can be tuned in practice. The authors state that, “It is worth noting that different tasks can have different values of α to capture the heterogeneity of dataset qualities across tasks” but in practice, choosing a separate α for each task seems undesirable. If α was tuned for this method but no hyperparameters were tuned for baselines, this would also be concerning as this would bias the results in favor of MerPO.

There are some unjustified, or at least confusing claims, such as:

“Because tasks are trained on offline datasets, value overestimation (Fujimoto et al., 2019) inevitably occurs in offline Meta-RL” Overestimation, as far as I know, is only an issue with value-based methods.

“we study a more general offline Meta-RL problem.” I do not see how the problem statement in this paper is more general than the offline meta-RL problem present in past papers.

“Learnt dynamics models not only serve as a natural remedy for task structure inference in offline Meta-RL, but also facilitate better exploration of out-of-distribution state-actions by generating synthetic rollouts” Evidence that learnt dynamics model help specifically because they facility better exploration is not provided.

“Our results also provide a guidance for the algorithm design in terms of how to appropriately select the weights in the interpolation” I do not see how the theoretical results can practically guide appropriate choosing alpha. In the end, it seems like alpha still had to be chosen empirically.

Lastly, I have a clarification question: Can the method be applied to new, non-offline meta-RL tasks? One limitation of the approach is that RAC seems to depend on having a behavior policy to regularize against, making it impossible to use the resulting policy for online meta-RL.

**Summary Of The Paper:**

The authors argue that offline meta-RL methods do not learn from poor data well, by demonstrating that COMBO (a single-task offline RL method) outperforms FOCAL (an offline meta-RL method) only when the training data is good. They use this to motivate a method called MerBO, which involves (1) learning a meta-dynamics model with proximal meta-RL and (2) updating a policy with real and synthetic data using a method called “RAC”, which is equivalent to COMBO but with an added KL regularizer against the behavior and meta-policy, and (3) updating the meta-policy used by RAC. The authors prove that under certain assumptions, the resulting policy will outperform both the meta-policy and the behavior policies. The authors compare the method to existing offline meta-RL methods and demonstrate that (1) the use of a dynamics model is important, (2) the addition of the behavior cloning regularizer is important, and (3) the use of the dynamics model is important to perform well on held-out offline RL tasks.

**Summary Of The Review:**

The paper does not provide compelling evidence that their method addressed a critical problem, and the experiments make it difficult to know if the method proposed really outperforms current meta-offline RL method since details on tuning and important comparisons were not included. Moreover, the overall method is rather complex and potentially limited to the pure-offline RL (unless I am mistaken) evaluation scenario, making the practicality of the method questionable.

Edit: Based on the rebuttal, I've increased my score up to a 6.

---

> ### Author Response · Authors · 2021-11-22
> **Reply to Reviewer xD1s (3/3)**
>
> Q7: I do not see how the theoretical results can practically guide appropriate choosing $\alpha$. In the end, it seems like $\alpha$ still had to be chosen empirically.
>
> A7: We have the following clarifications about the guidance of Theorem 1.
>
> - Theorem 1 provides a range of $\alpha$, e.g., smaller than 0.5, for which the safe policy improvement property can be achieved universally for any behavior policy and meta-policy. Following the widely-adopted strategy of learning conservatively in the literature of offline RL (see e.g., Fujimoto et al. 2019, Kumar et al. 2019, Kumar et al. 2020, Yu et al. 2020, Kidambi et al. 2020), a conservative strategy for a fixed $\alpha$ is to choose $\alpha$ to be close to 0.5. Moreover, we have also evaluated the performance for $\alpha$ larger than 0.5, and Figure 5 clearly shows that the performance degrades for such instances of $\alpha$.
>
> - Besides, Theorem 1 implies that the safe policy improvement can be achieved when the impacts of the meta-policy and the behavior policy are well balanced. This motivates us to adapt $\alpha$ in a certain range so as to balance i) the distance between the task-policy and the behavior policy and ii) the distance between the task-policy and the meta-policy.
>
>
> Q8: Can the method be applied to new, non-offline meta-RL tasks? One limitation is that RAC seems to depend on having a behavior policy to regularize against, making it impossible to use the resulting policy for online meta-RL.
>
> A8: We do not see a valid reason to apply an offline algorithm to an online RL setting, because the online algorithms with online sampling have much bigger advantage to outperform offline algorithms. The current design in the literature follow very different rationale for offline and online RLs, in order to respectively achieve good performance in each setting.
>
> Even regardless of the rationality mentioned above, we also kindly disagree with your claim that RAC cannot be applied in online tasks. Specifically, the behavior policy is only used to model the existing dataset. In the online setup, the dataset can be generated by the previous policies in the learning process, e.g., the replay buffer widely used in online RL. Correspondingly, the regularization based on the behavior policy in RAC becomes the regularization based on the previous policy, which is exactly the regularization used in TRPO (Schulman et al. 2015).

---

> > ### Comment · Reviewer_xD1s · 2021-11-26
> > **Re: Reply to Reviewer xD1s**
> >
> > Thank you for your clarifications.
> >
> > Q1: The motivation is rather unclear. The paper does not provide evidence that striking this balance is the main issue that must be addressed.
> >
> > Thank you for the additional experiments. However, this concern is still unaddressed. In particular, I do not understand this logical step
> >
> > > However, our Figures 1 and 11 show that such algorithms can perform poorly in the case when the behavior policy of the tasks are of good quality. This motivates the idea of balancing between staying close to meta-policy and to behavior policy.
> >
> > I agree that the method improves the performance. I also agree that Figure 1 and 11 demonstrates that COMBO outperforms FOCAL only when the training data is good.  However, the claim in the paper is that the method improves the performance _because_ it better strikes the right balance between exploring with the meta-policy and exploiting the offline dataset. But, again, I do not see direct evidence for this. Perhaps part of the difficulty is that it is unclear what it means to "strike the right balance." Perhaps the authors could devise an experiment that controls for the offline dataset while varying the exploration, or vice versa.
> >
> > > We emphasize, "motivation" by itself does not necessarily mean "anything affirmative"
> >
> > I do not understand this sentence.
> >
> > > but the main point of the paper is to show that our developed new algorithm MerPO that balances the tradeoff indeed improves upon existing approaches as shown in Figure 6 and 7
> >
> > I completely grant that MerPO improves over the existing approaches. What I take issue with is the explanation that it improves "by strike [sic] the right balance between exploring with the meta-policy and exploiting the offline dataset." Currently, I do not see evidence for this explanation, and I'm open to seeing what I am missing. However, as is, I think the paper would be better by either removing this claim or adjusting the statement to say that it is a speculation or untested hypothesis.
> >
> > Q2: Thank you for the response. I was mistaken and understated the improvement of past work.
> >
> > Q3: Thank you for adding the clarification around the adaptive alpha. This helps more clearly tie the work to the Theorem and address my concern around tuning. My remaining comment is:
> >
> > > We also kindly disagree that having being tuned yields bias in comparison with baselines. The hyperparameter  is part of the design of the new algorithm, and having it tuned to achieve better performance is fully valid.
> >
> > If tuning a hyperparameter were actually part of your algorithm, then the x-axis should reflect the number of trials needed to find the correct hyperparameter, but the adaptive $$\alpha$$ results make this concern moot.
> >
> > Q4: Thank you for the clarification.
> >
> > Q5: Thank you for the clarification. It would be good to include this in the paper.
> >
> > Q6: Perhaps the confusion is around the word "exploration." Typically, when I think of exploration, I think of the policy needing to explore a completely new part of the state space (E.g. a new room). Is there much of a difference between better exploration and better generalization?
> >
> > Q7: Thank you for the clarification.
> >
> > Q8: I see. Thank you.
> >
> > Overall, I've increased my score up to a 5. I still believe that the paper makes a large claim regarding _why_ the method improves performance that I do not see evidence for or understand.

---

> > > ### Author Response · Authors · 2021-11-28
> > > **Reply to Reviewer xD1s**
> > >
> > > Many thanks for your prompt response and further comments. We now have a better understanding of your concerns. Below are our responses.
> > >
> > > Q1: The claim in the paper is that the method improves the performance because it better strikes the right balance between exploring with the meta-policy and exploiting the offline dataset. But, I do not see direct evidence for this. Perhaps part of the difficulty is that it is unclear what it means to "strike the right balance."
> > >
> > > A1: Many thanks for further clarifying your comments. In what follows, we elaborate what we meant by "striking the right balance between exploring with the meta-policy and exploiting the offline dataset".
> > >
> > > - "Exploring with the meta-policy": In offline meta-RL, the meta-policy is learnt by leveraging offline datasets of "multiple" RL tasks. Thus, for a specific task, staying close to the meta-policy (i.e., using meta-policy as a regularizer) encourages the learnt policy to explore state-actions that are not seen in this specific task's dataset but have been encountered by some other tasks during the meta-learning.
> > >
> > >
> > > - "Exploiting the offline dataset": For each specific task, its offline dataset is collected by a corresponding behavior policy. "Exploiting the offline dataset" here means exploiting the existing state-actions in such a task-specific dataset, and hence the learned task-specific policy will stay close to the behavior policy. Here, the advantage is that the risk of making mistakes is reduced.
> > >
> > >
> > > - "Striking the right balance": In order to take the advantages of both of the above strategies, the design of our algorithm on the one hand applies the meta-policy to guide the task-specific policy to "explore" unseen state-actions in each task's dataset (which are encountered by meta-policy in other tasks), and on the other hand trains the task-specific policy to "exploit" this task's dataset, which thus stays close to the behavior policy and makes fewer mistakes. The tradeoff between the two is achieved in our algorithm by using a weight parameter $\alpha$ in the objective function to trade off between the two regularizer terms: one corresponding to meta-policy, and the other one corresponding to behavior policy.
> > >
> > > To make our statement more precise, we will change our statement to "balance the tradeoff between exploring the unseen state-actions by staying close to the meta-policy and exploiting the existing state-actions by staying close to the behavior policy". We will also further clarify our statement as we explain above in the final revision.
> > >
> > >
> > > Q3: If tuning a hyperparameter were actually part of your algorithm, then the x-axis should reflect the number of trials needed to find the correct hyperparameter, but the adaptive $\alpha$ results make this concern moot.
> > >
> > > A3: We have the following clarifications about $\alpha$.
> > >
> > > - For adaptive $\alpha$ setting, we start with an initial value $\alpha$ based on Theorem 1, and then at each iteration $k$ of algorithm, $\alpha_n^k$ is updated only once using one-step gradient descent with respect to Eq. (11) from $\alpha_n^{k-1}$ at the previous iteration $k-1$. Here, the automatic tuning process of $\alpha$ has already been included into the meta-training process. And Figure 6 on the convergence behavior has implicitly included the effort for adapting $\alpha$ during the learning process.
> > >
> > > - For fixed $\alpha$ setting, we only use grid search once to select $\alpha=0.4$ before all the trials in Figure 6, and generally apply this $\alpha$ to all the environments considered in the experiments. This selection procedure of a fixed parameter and the plot format in Figure 6 follow the previous studies, e.g., CQL, COMBO, FOCAL, all of which need to select certain fixed hyperparameters before the final trials and use the x-axis to show the number of iterations conducted in the final trials.
> > >
> > >
> > > Q6: Is there much of a difference between better exploration and better generalization?
> > >
> > > A6: We fully agree with the reviewer about the understanding of "exploration". The word "generalize" used in our previous response may have caused confusion, which should be "explore".
> > >
> > > Our general understanding about "exploration" versus "generalization" is as follows. "Exploration" usually means exploring the new state-actions during the learning process to obtain a desired policy, whereas "generalization" has a much more broad range of use contexts and can refer to the case of applying an already learnt policy to a new task (possibly under a different MDP). In this paper, in the context of this paper, we should use "exploration" to be more precise.

---

> > > > ### Comment · Reviewer_xD1s · 2021-11-30
> > > > **Thank you for the clarification**
> > > >
> > > > Thank you for the clarification. I have updated my score to reflect that my concerns have been addressed.

---

> > > > > ### Author Response · Authors · 2021-11-30
> > > > > **Thank you for your further updates!**
> > > > >
> > > > > We thank the reviewer very much for further reviewing our response and increasing the score!

---

> ### Author Response · Authors · 2021-11-22
> **Reply to Reviewer xD1s (2/3)**
>
> Q3: It is unclear how $\alpha$ is tuned or can be tuned in practice. In practice, choosing a separate $\alpha$ for each task seems undesirable. If $\alpha$ was tuned for this method but no hyperparameters were tuned for baselines, this would also be concerning as this would bias the results in favor of MerPO.
>
> A3: We have the following clarifications about $\alpha$.
>
> - We kindly disagree with your statement that "in practice, choosing a separate $\alpha$ for each task seems undesirable". As suggested by both Reviewer jFYQ and Reviewer zWVS, it is better to adaptively tune $\alpha$ within the learning process for each task, because the meta-policy changes in the learning process and the qualities of the behavior policies vary across different tasks. In fact, we did design a method called MerPO-Adp that adapts $\alpha$ across different tasks during the learning process, and have shown in experiments (see Figure 6 in Section 5.2) that such an algorithm improves over the vanilla MerPO with fixed $\alpha$.
>
> - We also kindly disagree that having $\alpha$ being tuned yields bias in comparison with baselines. The hyperparameter $\alpha$ is part of the design of the new algorithm, and having it tuned to achieve better performance is fully valid.
>
>
> Q4: Overestimation is only an issue with value-based methods.
>
> A4: Value overestimation is an important issue in dynamic programming methods when applied in offline RL, due to out-of-distribution queries in the offline setting (e.g., Fujimoto et al. 2019, Kumar et al. 2019). These methods includes not only value-based methods, but also the most effective off-policy policy gradient methods (e.g., Gelada \& Bellemare 2019, Nachum \& Dai 2020), which often require estimating the value function or the state-marginal density ratios via dynamic programming (see section 3.5 in Levine et al. 2020).
> We have revised the statement from "occurs in offline Meta-RL" to "occurs in dynamic programming based offline Meta-RL".
>
>
> Q5: The sentence from the paper that "We study a more general offline meta-RL problem" is unjustified. I do not see how the problem statement in this paper is more general than existing work.
>
> A5: We kindly disagree with the reviewer. As stated in the related work section, our work does not consider special scenarios or make special assumptions as in previous studies about offline meta-RL. To recap, we summarize a few related existing studies here:
>
> - Li et al. 2020a considers a special scenario where the task identify is spuriously inferred due to biased datasets, whereas we do not have such a requirement.
>
> - Dorfman \& Tamar 2020 assumes known reward functions and considers the offline dataset in the format of full trajectories, whereas we do not make such an assumption.
>
> - Mitchell et al. 2020 also considers the offline dataset in the format of full trajectories in order to evaluate the Monte Carlo return from a state in the dataset, whereas we do not have such a requirement.
>
> - Li et al. 2020b assumes that the MDP is deterministic, whereas we do not make such an assumption.
>
>
> Q6: Evidence that learnt dynamics model help specifically because they facilitate better exploration is not provided.
>
> A6: As evidenced in model-based offline RL approaches, e.g., MOPO (Yu et al. 2020), MoREL (Kidambi et al. 2020), COMBO (Yu et al. 2021b), model-based approaches can generalize beyond the state and action support of the offline dataset by generating synthetic data using the learnt dynamics model, and effectively solve the tasks by reaching to the states that are unseen in the offline dataset (see Fig 2 in Yu et al. 2020). In contrast, most model-free approaches, e.g., Fujimoto et al. 2019, Kumar et al. 2020, only consider states that lie in the offline dataset and do not consider the states that are out-of-distribution. Again, our statement aims to emphasize that the learnt dynamics model enables "exploration" of out-of-distribution state-actions, due to the capability of generating synthetic samples.

---

> ### Author Response · Authors · 2021-11-22
> **Reply to Reviewer xD1s (1/3)**
>
> Thank you for your detailed reviews and valuable comments. In our revision of the paper, we added new experiments on adaptive $\alpha$ in section 5.2 and on the comparison between FOCAL and COMBO in the appendix A.2.4, and made various revisions throughout the paper based on all reviewers’ comments. All our changes are highlighted with blue-colored texts. New comments on these changes are very welcome!
>
> Q1: The motivation is rather unclear. The paper does not provide evidence that striking this balance is the main issue that must be addressed.
>
> A1: We believe that the motivation of the proposed method has a reasonable basis and the experiment supports, and in fact it has been well received by the other three reviewers. We further clarify the reviewer's several concerns as follows.
>
> - We first clarify that the word "may" means "there exists non-trivial evidence" as we show in Figure 1, which we believe is a direct evidence.
>
> - Second, to provide more evidences that "existing offline Meta-RL fails to generalize equally well...", we have conducted more experiments on other environments as shown in Figure 11 in Appendix A.2.4. The performance comparison further confirms the evidence in Figure 1. To be more precise, we have also changed our statement from "fails" to "fails in several standard environments".
>
> - Finally, for the reviewer's main concern about "the motivation and evidence that striking this balance is the main issue", we clarify our logic as follows. Existing studies in offline meta-RL fully use a meta-policy to learn task-specific policies without consideration of behavior policy of these tasks. However, our Figures 1 and 11 show that such algorithms can perform poorly in the case when the behavior policy of the tasks are of good quality. This motivates the idea of balancing between staying close to meta-policy and to behavior policy. We emphasize, "motivation" by itself does not necessarily mean "anything affirmative"; but the main point of the paper is to show that our developed new algorithm MerPO that balances the tradeoff indeed improves upon existing approaches as shown in Figure 6 and 7, which closes our loop from "motivation" to "technical affirmation"!
>
>
> Q2: The paper is a relatively complex combination of existing components (COMBO, Proximal meta-RL, behavior cloning regularization), and so the onus is on the paper to demonstrate exceedingly good results to justify the complexity of the method. The method only mildly improves over FOCAL, and the paper would be strengthened by comparing to MACAW and BORel.
>
> A2: We kindly disagree with you that our approach is a relatively complex combination of existing components and only improves mildly over FOCAL.
>
> - Regarding the complexity, our approach is in the same spirit as in general model-based approaches (and hence not more complex than these approaches), which include i) learning a dynamics model and 2) using a model-free approach to learn the policy based on existing data and model generated rollouts.
>
> - Our approach is not a combination of existing components. A novel regularized within-task policy improvement approach is devised to calibrate the tradeoff between exploiting the offline dataset and following the meta-policy, through a weighted interpolation between two different regularizers. This weighted interpolation (not just the behavior regularization) is the foundation to establish the safe policy improvement property over both the behavior policy and the meta-policy, which indeed serves as a key building block of our offline meta-RL approach MerPO.
>
> - Our algorithm improves over FOCAL "substantially", not "mildly". Compared to FOCAL which demonstrates the state-of-the-art performance, our approach achieves around 21\% improvement in Walker-2D-Params, 35\% improvement in Half-Cheetah-Fwd-Back, and 100\% improvement in Ant-Fwd-Back. We would appreciate that if the reviewer can explain more clearly what performance would match what the reviewer requests as "exceedingly good results".
>
> - We didn't compare MerPO with MACAW and BORel, because i) MACAW considers the offline dataset in the format of full trajectories, so as to evaluate the Monte Carlo return from a state in the dataset and then the advantage of a particular action over the learnt value function of the state. In contrast, we consider a more general setup of the offline dataset containing individual transition tuples (see e.g., "Critic Regularized Regression" (Wang et al. 2020), "Continuous Doubly Constrained Batch Reinforcement Learning" (Fakoor et al. 2021)), where it is infeasible to evaluate the Monte Carlo return from a state. ii) BORel assumes known reward functions for the reward relabelling; whereas we consider a more practical scenario where the reward functions are unknown, which is particularly important for the meta environments such as Half-Cheetah-Fwd-Back and Ant-Fwd-Back (where all tasks have different reward functions).

---

### Official Review · Reviewer_i86B · 2021-11-02

**Correctness:** 4
**Technical Novelty And Significance:** 3
**Empirical Novelty And Significance:** 3
**Recommendation:** 6
**Confidence:** 4

**Main Review:**

This paper is motivated by what the authors refer to as an inherent trade-off between exploiting the data in the offline dataset, and exploring out-of-distribution states (by means of a model). Empirically, they motivate this trade-off by noting that the quality of the data-set has a large impact on the efficacy of offline meta-RL approaches, and that single-task approaches to offline RL can be superior if data-quality is high.

To allow out-of-distribution exploration, they meta-learn an initialisation for a state-transition model via standard supervised meta-learning approaches. This initialisation is then used to rapidly adapt the model to a given dataset. The main part of the paper revolves around how to meta-learn a policy that can act as a regularizer in a policy-improvement step. They demonstrate empirically that if task-adaptation is regularised only towards the meta-learned policy, then performance can be surprisingly poor in the even that the meta-policy does not correspond to useful behaviour on the new task. Instead, the authors propose to regularize policy-improvement updates both towards the meta-learned policy and towards the behaviour policy implicitly defined in the offline dataset.

While the authors make a good case for their design choices, my main issue with this paper is that it complicates the presentation unnecessarily, which obfuscates connections to prior works. As far as I can tell, the latter regularizer (towards the behaviour policy) corresponds to COMBO, but this is not made clear until several pages later in the experimental section. Hence, the gist of this paper is to extend COMBO to the multi-task setting by meta-learning, while also meta-learning a prior policy for proximal policy updates. This could be made much clearer, which would not only help place the contribution in proper context but also make the paper much more readable.

A main strength of this paper is the careful motivation of each component of the proposed method. The authors demonstrate theoretically that both these regularisers are important for policy improvement, clearly motivating the need for a meta-learned prior as well as regularisation towards the dataset's behaviour policy. This observation holds also empirically, as the authors demonstrate that their method is at least as good as using only one of the two regularisers under various assumptions of data-quality. They also demonstrate that learning a model is critical for performance, and that their proposed method is better or on par with established offline meta-RL baselines.

Overall, while I think the presentation of the method can be made much simpler and clearer, I believe this paper presents interesting findings for offline RL and has a strong proposal for an offline meta-RL algorithm.

**Summary Of The Paper:**

This paper proposes a model-based meta-RL approach to offline learning over a distribution of MDPs (offline meta-RL). The proposed method is inspired by COMBO and in fact is a direct extension of it to the multi-task setting. The extension involves meta-learning a model-initialisation (or prior) that can be rapidly adapted to new offline RL tasks. They further add a proximal RL policy-improvement operator, where the prior policy is meta-learned over the task distribution. They establish theoretical guarantees for their method and conduct a careful empirical investigation to motivate its design, while establishing that it provides additional benefits compared to relevant baselines.

**Summary Of The Review:**

I recommend acceptance of this paper. Main strengths of the paper are:

+ Compelling motivation of algorithm
+ Careful analysis of its design
+ Competitive performance

Main weaknesses are
- Overly complicated presentation of the algorithm

Post rebuttal:

I've read other reviews and the author's rebuttal and maintain my recommendation.

---

> ### Author Response · Authors · 2021-11-22
> **Reply to Reviewer i86B**
>
> Thank you for your thorough reviews and constructive comments. In our revision of the paper, we added new experiments on adaptive $\alpha$ in section 5.2 and on the comparison between FOCAL and COMBO in the appendix A.2.4, and made various revisions throughout the paper based on all reviewers’ comments. All our changes are highlighted with blue-colored texts. New comments on these changes are very welcome!
>
> Q: The paper complicates the presentation unnecessarily, which obfuscates connections to prior works.
>
> A. Thanks for the comments. As suggested by the reviewer, our revision of the paper clarified the connections with COMBO in the introduction, Section 4.2.1 and Section 5.2 (ablation study). Intuitively, RAC generalizes COMBO to the multi-task setting, while introducing a novel regularized policy improvement module to strike a right balance between the impacts of the meta-policy and the behavior policy.

---

### Official Review · Reviewer_jFYQ · 2021-11-02

**Correctness:** 4
**Technical Novelty And Significance:** 2
**Empirical Novelty And Significance:** 2
**Recommendation:** 6
**Confidence:** 4

**Main Review:**

Advantage:
The writing of the paper is clear and easy to follow. The proposed MerPO outperforms the state-of-art baselines and is intuitively explained well. The experiments show most of the performance of the components in the proposed algorithms, model-based vs model-free, involving behavior policy or not, the influence of the alpha, etc.

Disadvantage/Problems under concern:
1. About the important regularization parameter $\alpha$. 1) Will an adaptive $\alpha_t$ be better? Since the quality of the meta-policy $\pi_c$ will be different during the learning process. 2) Do we need to use different alpha for each task when the behavior policy property of each task is different? This scenario seems didn't been discussed enough?
2. How to choose alpha without information on the performance of the behavior policy and meta-policy? Do we need to try it out?
3. There are some undefined or unclear notations, such as what is $\rho_n$ in equation (5).




**Summary Of The Paper:**

Targeting offline meta-reinforcement learning, the work proposes a model-based method called MerPO, with conservative value evaluation and individual policy improvement method with the tradeoff between the meta-policy and the behavior policy influence.

The main algorithm MerPO includes an initialization step and a two-loops meta-learning approach. The initialization step learns the model of the meta-model and dynamics for each task. With the fixed estimations of the models of tasks, the proposed merPO will alternatively update the policies for each task and the meta-policy.

The main contribution claimed in this paper is that the proposed method is a more robust method with the design of a regularization term involving the behavior policy, which improves the meta-learning policy when the behavior policy is actually better than the current meta-policy. Theoretical guarantees are displayed and experiments are conducted to show the performance of MerPO.

**Summary Of The Review:**

The main contribution of this paper is well claimed and verified. With a small revision, the algorithm can achieve better performance on the examined tasks in this paper.

---

> ### Author Response · Authors · 2021-11-22
> **Reply to Reviewer jFYQ**
>
> Thank you for your thorough reviews and constructive comments. In our revision of the paper, we added new experiments on adaptive $\alpha$ in section 5.2 and on the comparison between FOCAL and COMBO in the appendix A.2.4, and made various revisions throughout the paper based on all reviewers’ comments. All our changes are highlighted with blue-colored texts. New comments on these changes are very welcome!
>
>
> Q1: About $\alpha$. 1) Will an adaptive $\alpha_t$ be better? Since the quality of the meta-policy $\pi_c$ will be different during the learning process. 2) Do we need to use different $\alpha$ for each task when the behavior policy property of each task is different?
>
> A1: Many thanks for the insightful comments. Following the reviewer's comment, we designed an algorithm with $\alpha$ adaptive to tasks and during the learning process, and experimentally showed that such an algorithm does outperform our previous algorithm with fixed $\alpha$. We summarize our result as follows.
>
> As suggested by the reviewer, since the meta-policy $\pi_c$ changes during the learning process and the qualities of the behavior policies vary across different tasks, we adapt $\alpha$ across different iterations and tasks so as to achieve a 'local' balance between the impacts of the meta-policy and the behavior policy. We name the new method as MerPO-Adp. Our design idea obeys the widely-adopted rule of learning conservatively in offline RL (see e.g., Fujimoto et al. 2019, Kumar et al. 2019, Kumar et al. 2020, Yu et al. 2020, Kidambi et al. 2020). Specifically, at each iteration $k$, given the task-policy $\pi_n^k$ for task $n$ and the meta-policy $\pi_c^k$ at iteration $k$, we update $\alpha^k_n$ using one-step gradient descent for the following optimization problem:
>
> $$\min_{\alpha^k_n}~ (1-\alpha^k_n)[D(\pi^k_{n},\pi_{\beta,n})-D(\pi^k_n,\pi^k_c)], s.t.~ \alpha^k_n\in[0.1,0.5].$$
>
> The basic idea is to adapt $\alpha_n^k$ in order to strike a balance between $D(\pi^k_n,\pi_{\beta,n})$ and $D(\pi^k_n,\pi^k_c)$, based on the implication of Theorem 1  that the safe policy improvement can be achieved when the impacts of the meta-policy and the behavior policy are well balanced for offline RL. Specifically, at iteration $k$ for each task $n$,  $\alpha^k_n$ is increased when the task-policy $\pi_n^k$ is closer to the meta-policy $\pi^k_c$, and is decreased  when the task-policy is closer to the behavior policy. Further,  note that $\alpha^k_n$ is constrained in the range $[0.1,0.5]$ as suggested by Theorem 1.
>
> We conduct experiments to evaluate the performance of the new algorithm MerPO-Adp in Section 5.2. As expected, Figure 6 indicates that MerPO-Adp can achieve better or comparable performance compared to MerPO with a fixed $\alpha$.
>
>
> Q2: How to choose $\alpha$ without information on the performance of the behavior policy and meta-policy? Do we need to try it out?
>
> A2: As mentioned by the reviewer, one key challenge to choose $\alpha$ is that the qualities of both the behavior policy and the meta-policy are unknown. To overcome this difficulty, we  follow the widely-adopted conservative learning philosophy in the literature of offline RL to choose $\alpha$ in the range suggested by Theorem 1. Particularly, i) for a fixed $\alpha$,  we select it to be closer to 0.5; ii) for an adaptive $\alpha$, we develop the algorithm MerPO-Adp as explained in A1, which adapts $\alpha_n^k$ in order to balance between $D(\pi^k_n,\pi_{\beta,n})$ and $D(\pi^k_n,\pi^k_c)$ at each step.
>
>
> Q3: There are some undefined or unclear notations, such as what is $\rho_n$ in equation 5.
>
> A3: Many thanks for pointing out the unclear notations. Here $\rho_n(s)$ is the state marginal of $\rho_n(s,a)$ for task $n$, where $\rho_n(s,a)$ is the discounted marginal state-action distribution when rolling policy $\pi$ in the learnt MDP. We have fixed these issues in the revision.

---

> > ### Author Response · Authors · 2021-11-28
> > **Reply to Reviewer jFYQ**
> >
> > We sincerely thank the reviewer again for the thorough reviews and constructive comments! Since the final stage of the discussion will end soon, please let us know if you have further questions on our response, and we will be more than happy to answer your questions.

---

> > > ### Comment · Reviewer_jFYQ · 2021-11-29
> > > **Response to author**
> > >
> > > Thanks for all your answers to my questions. My concerns have been addressed and I'd like to raise my score to 6.

---

> > > > ### Author Response · Authors · 2021-11-29
> > > > **Many thanks for your further updates!**
> > > >
> > > > We thank the reviewer very much for further reviewing our response and increasing the score!

---

### Decision · Program_Chairs · 2022-01-20

**Decision:**

Accept (Poster)

**Comment:**

**Summary**

This paper proposes a novel offline model-based meta-RL approach called MerPO. MerPO combines conservative value iteration with proximal RL policy iteration.  The proposed method is novel despite having some similarities to approaches like COMBO. The paper compares against it in the experiments. The paper provides both empirical and theoretical justification for the proposed approach.

**Final Thoughts**

Overall, I think the authors did a pretty good job at addressing the reviewers' concerns. Overall, I think this is an interesting contribution to the ICLR community. The reviewers were all positive about this paper. For the camera-ready version of the paper, I would recommend the authors to go over the reviewers' concerns again and make sure that those concerns are addressed in the paper too as they did in the rebuttal. Some captions are pretty short; for example, see the captions of figure 6 and figure 7. I would recommend the authors add more description to the captions in the camera-ready version of this paper.